# CAN LLMS GENERATE DIVERSE MOLECULES? TOWARDS ALIGNMENT WITH STRUCTURAL DIVERSITY

## ABSTRACT

Recent advancements in large language models (LLMs) have demonstrated impressive performance in generating molecular structures as drug candidates, which offers significant potential to accelerate drug discovery. However, the current LLMs overlook a critical requirement for drug discovery: proposing a diverse set of molecules. This diversity is essential for improving the chances of finding a viable drug, as it provides alternative molecules that may succeed where others fail in wet-lab or clinical validations. Despite such a need for diversity, the LLMs often output structurally similar molecules from a given prompt. While decoding schemes like beam search may enhance textual diversity, this often does not align with molecular structural diversity. In response, we propose a new method for fine-tuning molecular generative LLMs to *autoregressively generate a set of structurally diverse molecules*, where each molecule is generated by conditioning on the previously generated molecules. Our approach consists of two stages: (1) supervised fine-tuning to adapt LLMs to autoregressively generate molecules in a sequence and (2) reinforcement learning to maximize structural diversity within the generated molecules. Our experiments show that (1) our fine-tuning approach enables the LLMs to better discover diverse molecules compared to existing decoding schemes and (2) our fine-tuned model outperforms other representative LLMs in generating diverse molecules, including the ones fine-tuned on chemical domains.

## 1 INTRODUCTION

Recent advances in large language models (LLMs) have demonstrated remarkable potential to accelerate scientific discovery by leveraging their language processing capabilities. This progress has been particularly impactful for candidate design problems such as drug discovery (Pei et al., 2024), protein sequence design (Zhuo et al., 2024), and material design (Gruver et al., 2024). In particular, with expansive biomolecular datasets and molecular string representations, e.g., SMILES (Weininger, 1988) or SELFIES (Krenn et al., 2020), LLMs have demonstrated impressive abilities to generate molecules from textual descriptions, e.g., molecular properties (Edwards et al., 2022; Ye et al., 2023; Pei et al., 2024), as illustrated in Figure 1.

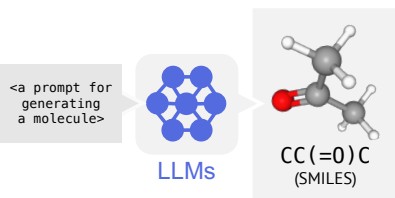

Figure 1: **LLMs generate a molecule with SMILES representation.**

However, current LLM-based molecular generation approaches (Edwards et al., 2022; Ye et al., 2023; Pei et al., 2024) often overlook a critical requirement for drug discovery: *proposing a diverse set of molecules*. In computer-aided drug discovery pipelines, identifying a single molecule with a desired property does not guarantee success in real-world pipelines that require additional cell-based studies and clinical trials (Vamathevan et al., 2019). Therefore, drug discovery requires a collection of structurally diverse molecules, as illustrated in Figure 2.[1] The generation of structurally diverse molecules increases the chances of finding a viable drug candidate (Xie et al., 2023), as different molecules may succeed where others fail. This diversity is essential to enhance the robustness and success of the drug discovery process (Krantz, 1998; Hong et al., 2020; Sadybekov & Katritch, 2023).

---

[1]The diversity of molecules is evaluated with structural features, e.g., the presence of specific substructures.

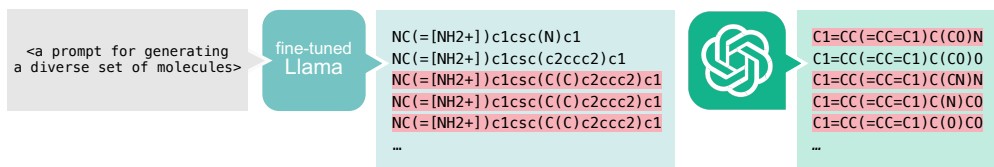

Figure 2: **The compute-aided drug design.** A collection of structurally diverse molecules is required to increase the chance of identifying viable drug candidates in the real world.

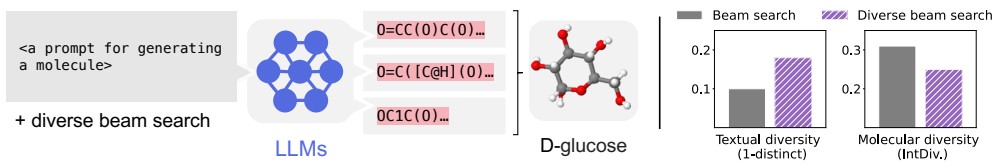

(a) Existing LLMs (Ye et al., 2023; OpenAI, 2023) lack the ability to generate a diverse set of molecules.

(b) **(Left)** Diverse output sequences (SMILES) induce the same molecular structures. **(Right)** Improved textual diversity via diverse beam search does not enhance molecular diversity in the experiments (Section 4.1).

Figure 3: **Existing works on LLMs fail to generate diverse molecules.** The existing decoding schemes (Vijayakumar et al., 2018) for diverse sequence generation and LLMs for chemical tasks fail to capture the molecular diversity, and may induce structurally identical molecules.

In response, we explore the use of LLMs for diverse molecular generation. We begin by identifying the limitations of recent LLMs (Ye et al., 2023; OpenAI, 2023) and decoding schemes (Vijayakumar et al., 2018; Su et al., 2022) in generating diverse molecules. Then, we present a new method for fine-tuning LLMs to generate diverse molecules. Our approach can be broadly applied to other LLM-based candidate design problems, e.g., computer-aided design (Wu et al., 2023a).

**Existing LLMs have limitations in generating diverse molecules.** To obtain diverse molecules, one may consider querying the recently developed generalist LLMs, e.g., Llama (Touvron et al., 2023) or ChatGPT (OpenAI, 2023). However, our empirical observation in Figure 3(a) reveals that even the most recent models produce structurally identical or highly similar molecules from the given prompt.[2] This observation aligns with previous observations that have shown LLMs may fail to generate diverse outputs (Kirk et al., 2024) for general text-based domains.

**Decoding schemes for diversified generation do not align with molecular diversity.** We also acknowledge the existence of decoding schemes, e.g., diverse beam search (Vijayakumar et al., 2018) or contrastive beam search (Su et al., 2022), which have been proposed to improve the diversity of output sequences generated by LLMs. However, these decoding schemes are limited to improving the textual diversity which often does not correspond to molecular structural diversity, e.g., there exist many SMILES or SELFIES strings that correspond to the same molecule, as illustrated in Figure 3(b).

**Our approach.** We repurpose existing molecular generative LLMs to autoregressively generate a diverse set of molecules from a single prompt. By enabling the LLMs to generate a new molecule conditioned on previously generated molecules, we expect the LLMs to learn to enhance the structural diversity between the generated molecules. To this end, we propose a two-stage approach to fine-tune LLMs: (a) a supervised fine-tuning stage to repurpose LLMs to autoregressively generate a sequence of multiple molecules and (b) a reinforcement learning stage to maximize the molecular structural diversity. Note that both stages do not require external diverse molecular datasets and are purely based on self-improvement procedures, where the LLMs train on the samples generated by themselves.

---

[2]ChatGPT-4o (OpenAI, 2023) generates different SMILES strings that map to an identical molecule.

In the supervised training stage, we train the LLMs to autoregressively generate a set of molecules in a single sequence. The training dataset, i.e., a set of molecules, is collected from the LLMs themselves through iterative sampling, and then filtered to enhance the quality, e.g., removing invalid molecules. However, this stage does not necessarily incorporate molecular diversity, as the training may not involve sufficiently distinct molecules (e.g., limitations in Figure 3(b)). To tackle this, we subsequently apply reinforcement learning with exploration towards discovering diverse molecules.

Next, in the reinforcement learning stage, we train LLMs to maximize the diversity of molecules within a generated sequence. However, for our task, conventional sequence-wise reinforcement learning (Ouyang et al., 2022) suffers from the credit assignment problem (Zhou et al., 2024): the challenges in identifying and promoting the generation of molecules responsible for increasing diversity, among a larger set of molecules in the sequence. To resolve this issue, we solve multi-stage molecule generation problems for a sequence of molecules, where the generation of each molecule aims to maximize the diversity with respect to the previously generated molecules. We train LLMs to maximize the associated rewards using proximal policy optimization (Schulman et al., 2017).

We compare our method with the decoding schemes for diversified generation (Vijayakumar et al., 2016; Su et al., 2022) and other representative LLMs, including chemical-task specialists (Edwards et al., 2022; Christofidellis et al., 2023; Pei et al., 2023; 2024), fine-tuned generalists on chemical domains (Fang et al., 2024; Yu et al., 2024), and the ChatGPT series (OpenAI, 2023; 2024). We observe that (1) our fine-tuning approach enables LLMs to better discover diverse molecules compared to existing decoding schemes and (2) our fine-tuned LLM outperforms other recent LLMs.

To conclude, our contributions can be summarized as follows:

- We are the first to explore the use of LLMs for generating diverse molecules.
- We first propose a fine-tuning approach for LLMs to generate diverse solutions, which presents a new direction distinct from existing approaches focused on the decoding scheme.
- Experimentally, our method outperforms the baselines in generating diverse molecular structures.

## 2 RELATED WORK

**Large language models (LLMs) for molecular generation.** Recent advancements in LLMs have shown increasing promise in scientific applications, especially for molecular generation (Edwards et al., 2022; Pei et al., 2023; Fang et al., 2024; Pei et al., 2024). First, Edwards et al. (2022) proposed MolT5 which translates between SMILES (Weininger, 1988) and molecular text descriptions. Text+Chem T5 (Christofidellis et al., 2023) is a model pre-trained on both the chemical and natural language domains. Next, BioT5 (Pei et al., 2023) considers T5 models pre-trained on datasets including bio-text, protein sequences, and molecules. Additionally, Ye et al. (2023), Fang et al. (2024), and Yu et al. (2024) fine-tuned generalist LLMs, e.g., Llama (Touvron et al., 2023), through biological instructions, molecular modifications, and large-scale molecular datasets, respectively.

**Decoding schemes for generating diverse output sequences.** To generate diverse and high-quality solution candidates from LLMs, existing literature on LLMs has studied improving decoding schemes. To acquire multiple distinct sequences with high likelihoods, one can consider employing beam search, which jointly decodes multiple distinct outputs (Och, 2003). To enhance diversity between sequences, Vijayakumar et al. (2016; 2018) incorporated token-wise differences between generated sequences in the beam search. Furthermore, Su et al. (2022) considered the contrast between the candidate sequences. In addition, Holtzman et al. (2020) proposed nucleus sampling, which enhances random sampling by balancing the quality and the diversity.

**Reinforcement learning (RL) for fine-tuning LLMs.** RL has been effectively applied to fine-tune LLMs, aligning them with desired behaviors expressed through reward signals. One notable example is RL from human feedback to align LLMs with human preference (Ouyang et al., 2022). In addition, there has been a surge in research on devising RL for LLMs as well, such as addressing multi-turn settings (Shani et al., 2024) and incorporating multiple fine-grained reward signals (Wu et al., 2023b). For molecular generation, Ghugare et al. (2024) proposed RL-based fine-tuning to generate a molecule satisfying target properties. However, to the best of our knowledge, there exist no prior RL-based approaches that aim to increase the diversity of LLM-generated outputs.[3]

---

[3]Additional related works, e.g., RL for diverse molecular generation and diversity metrics, are in Appendix A.

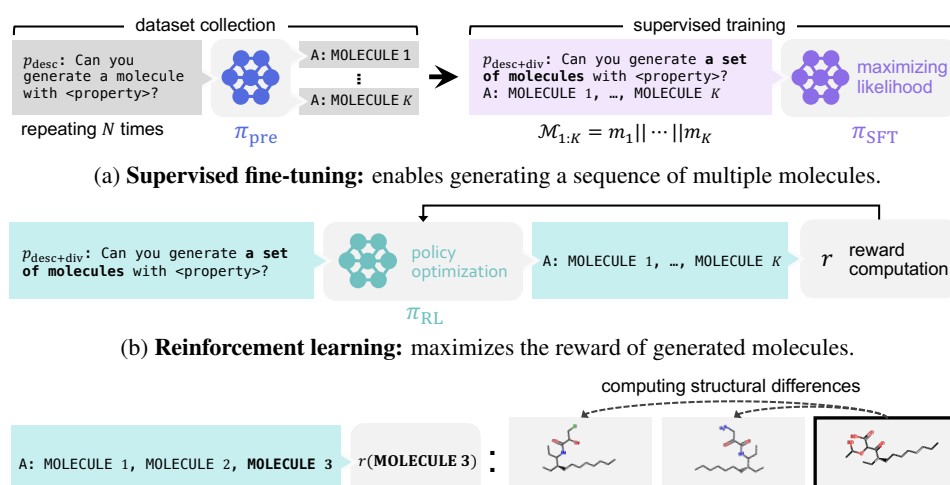

(a) **Supervised fine-tuning:** enables generating a sequence of multiple molecules.

(b) **Reinforcement learning:** maximizes the reward of generated molecules.

(c) **Reward computation:** captures the structural diversity.

Figure 4: **Illustration of proposed fine-tuning approaches.** We consider two stages for fine-tuning LLMs: a supervised fine-tuning Figure 4(a) and a reinforcement learning Figure 4(b). The prompts are simplified for explanatory purposes, and the actual prompts are provided in Appendix C.

## 3 METHOD

In this section, we present our method for fine-tuning LLMs to generate diverse molecules. Specifically, we consider applying fine-tuning to existing molecular generative LLMs that produce molecular representations such as SMILES or SELFIES. Importantly, our approach is versatile and can be broadly applied to other domains, e.g., protein sequence (Zhuo et al., 2024) or computer-aided design (Wu et al., 2023a). Furthermore, it leverages self-improvement techniques and does not require additional datasets containing diverse molecules.

**Overview.** Our goal is to generate a sequence of structurally diverse molecules from a given prompt by producing them in a single concatenated output. To achieve this, we fine-tune the LLMs in two stages: (a) a supervised fine-tuning phase that repurposes the LLMs to generate a sequence of molecules rather than a single one, and (b) a reinforcement learning phase aimed at further enhancing the structural diversity among the generated molecules.

**Task details.** In detail, we consider generating molecules from a prompt $p_{\text{desc}}$, where the prompt describes a molecular property that the generated molecules should possess. In this setting, we aim to generate diverse molecules that satisfy the given description $p_{\text{desc}}$. Here, the diversity is evaluated using similarity measures between the structural features of the molecules, e.g., the presence of specific atoms, or substructures (Bajusz et al., 2015).[4] We let $\mathcal{P}$ denote the prompts used for training.

### 3.1 SUPERVISED FINE-TUNING

We first describe our supervised fine-tuning process for repurposing the pre-trained LLMs to autoregressively generate multiple molecules in a sequence. This involves collecting a dataset of molecules from a pre-trained LLM $\pi_{\text{pre}}$, and then fine-tuning the LLM $\pi_{\text{SFT}}$ on the collected dataset. We describe the process in Figure 4(a) and Algorithm 1.

**Dataset collection.** The supervised training process is conducted with a set of training prompts $\mathcal{P}$. Initially, the pre-trained LLM $\pi_{\text{pre}}$ produces a set of

**Algorithm 1** Supervised fine-tuning

1: Initialize $\pi_{\text{SFT}}$ with $\pi_{\text{pre}}$
2: **repeat**
3:      Sample prompt $p_{\text{desc}} \sim \mathcal{P}$
4:      Sample $\{m_i\}_{i=1}^T$ from $\pi_{\text{pre}}(m \mid p_{\text{desc}})$
5:      Update $\{m_i\}_{i=1}^K \leftarrow \text{Filter}(\{m_i\}_{i=1}^T)$
6:      Maximize Equation (1) with $\{m_i\}_{i=1}^K$
7: **until** Converged
8: **Output:** fine-tuned $\pi_{\text{SFT}}$

---

[4]In Appendices A.2 and A.3, we discuss (1) detailed similarity measures for evaluating whether two molecules are similar or distinct and (2) detailed diversity metrics for evaluating a set of molecules, respectively.

molecules by iterative sampling molecules for a given prompt $p_{\text{desc}} \in \mathcal{P}$ as follows:

$$m_i \sim \pi_{\text{pre}} (m_i | p_{\text{desc}}) \quad \text{for } i = 1, \ldots, T,$$

where $m_i$ denotes the string representation of the molecule. In practice, we employ beam search to collect the set of molecules $\{m_i\}_{i=1}^T$. Then, we filter out the invalid string representations, duplicate molecules, and molecules that do not satisfy the given prompt $p_{\text{desc}}$. This results in reducing the set of molecules from $\{m_i\}_{i=1}^T$ to $\{m_i\}_{i=1}^K$. The details are described in Appendix B.

**Supervised training.** Given the filtered set of molecules $\{m_i\}_{i=1}^K$, we train the LLM $\pi_{\text{SFT}}$, which is initialized with $\pi_{\text{pre}}$, to generate them as a single concatenated sequence. We denote this sequence by $\mathcal{M}_{1:K} = m_1 || \cdots || m_K$, where $||$ denotes the concatenation of the molecular string representations. Specifically, given a modified prompt $p_{\text{desc+div}}$ describing the target property with a request for generating diverse molecules, we train the LLM to maximize the log-likelihood:

$$\log \pi_{\text{SFT}} (\mathcal{M}_{1:K} \mid p_{\text{desc+div}}). \tag{1}$$

However, the policy $\pi_{\text{SFT}}$ does not necessarily incorporate a molecular structural diversity, as the set of molecules $\{m_i\}_{i=1}^K$ collected from $\pi_{\text{pre}}$ may insufficiently involve diverse molecular structures (e.g., due to limitations in Figure 3(b)). To tackle this, we next introduce an online reinforcement learning stage with exploration towards discovering diverse molecules.

## 3.2 REINFORCEMENT LEARNING

We apply reinforcement learning to maximize the diversity of the generated molecules within a sequence. However, when applied to a sequence of molecules $\mathcal{M}_{1:K}$, conventional sequence-wise reinforcement learning (Ouyang et al., 2022) suffers from the credit assignment problem (Zhou et al., 2024): the challenge in identifying and promoting the generation of molecules responsible for increasing diversity, among a set of molecules $\{m_i\}_{i=1}^K$. To circumvent this, we introduce a molecule-wise reinforcement learning.[5]

| **Algorithm 2** Multi-stage RL fine-tuning |
| :--- |
| 1: Sample $p_{\text{desc}} \sim \mathcal{P}$ |
| 2: Sample $m_1 \sim \pi_{\text{RL}}(m_1 \mid p_{\text{desc+div}})$ |
| 3: Update $\pi_{\text{RL}}$ with PPO to maximize $r(m_1)$ |
| 4: **for** $k = 2, \ldots, K$ **do** |
| 5:     Sample $m_k \sim \pi_{\text{RL}}(m_k \mid \mathcal{M}_{1:k-1}, p_{\text{desc+div}})$ |
| 6:     Update $\pi_{\text{RL}}$ with PPO to maximize $r(m_k)$ |
| 7: **end for** |

Specifically, we consider reinforcement learning on a sequence of molecules $\mathcal{M}_{1:K}$ as learning in $K$ individual stages. Each stage corresponds to generating a molecule $m_k$ conditioned on a sequence of previously generated molecules $\mathcal{M}_{1:k-1}$. Then, the LLM $\pi_{\text{RL}}$ is trained to maximize the return of each stage, which is defined by the reward of the generated molecule $m_k$. The reward is defined as the diversity between the previously generated molecules $\{m_i\}_{i=1}^{k-1}$ and the new molecule $m_k$. We also incorporate an auxiliary reward to ensure that the molecule $m_k$ satisfies the given description $p_{\text{desc}}$. We illustrate our approach in Figure 4(b) and provide the detailed algorithm in Algorithm 2.

**Reward.** The reward evaluates the generated molecule $m_k$ with a diversity reward $r_{\text{div}}(m_k, \{m_i\}_{i=1}^{k-1})$ and a description-matching reward $r_{\text{match}}(m_k, p_{\text{desc}})$, as follows:

$$r(m_k) = \lambda_{\text{div}} r_{\text{div}}(m_k, \{m_i\}_{i=1}^{k-1}) + \lambda_{\text{match}} r_{\text{match}}(m_k, p_{\text{desc}}),$$

where the diversity reward $r_{\text{div}}$ evaluates structural differences between the molecule $m_k$ and the previously generated molecules $\{m_i\}_{i=1}^{k-1}$ by assessing their true molecular structures. Note that $r_{\text{div}}(m_1)$ is zero. The description-matching reward $r_{\text{match}}$ evaluates whether the molecule $m_k$ satisfies the given description $p_{\text{desc}}$. In a practical implementation, the final action that completes the molecule $m_k$ yields the reward $r(m_k)$. We set both reward coefficients $\lambda_{\text{div}}$ and $\lambda_{\text{match}}$ to one in our experiments. The detailed reward implementation is described in Appendix B.

**Policy optimization.** We optimize the LLM $\pi_{\text{RL}}$ to maximize the reward using proximal policy optimization (PPO; Schulman et al., 2017; Ouyang et al., 2022). Note that $\pi_{\text{RL}}$ is initialized with $\pi_{\text{SFT}}$. In addition, we combine per-token KL penalty from the supervised fine-tuned model at each token following prior studies (Ouyang et al., 2022).

---

[5]We compare both approaches in Table 4 of Section 4.4.

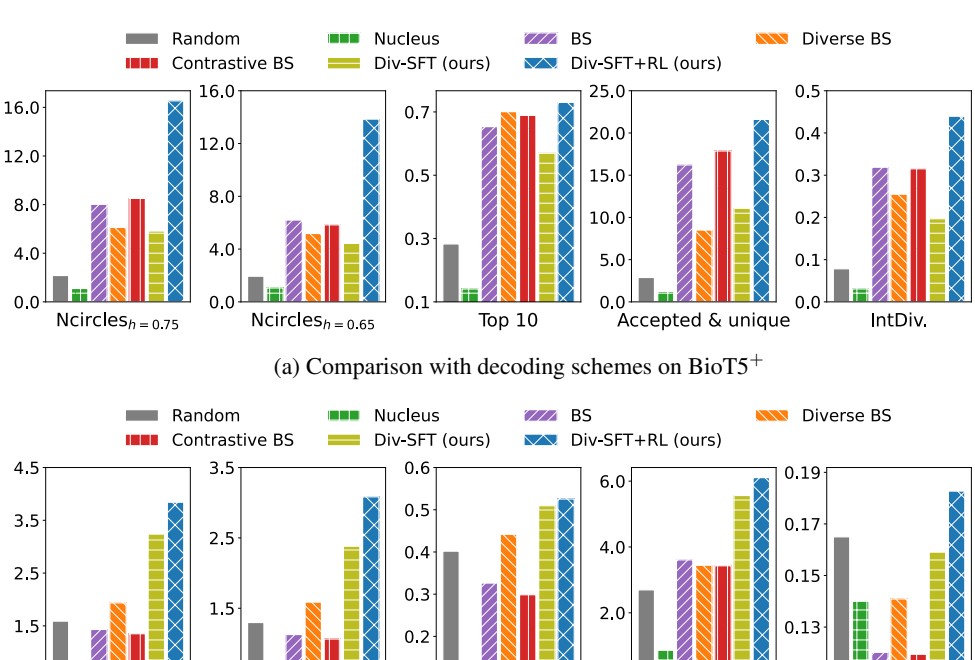

(a) Comparison with decoding schemes on BioT5$^+$

(b) Comparison with decoding schemes on MolT5

Figure 5: **Comparison with decoding schemes. NCircles** represents both quality and diversity-related metric. **Top 10 and Accepted & unique** represent quality-related metrics. **IntDiv.** represents a diversity-related metric. Our method generates more diverse and high-quality molecules compared to the baselines. Notably, our method makes a larger gap over the baselines on metrics related to capturing both quality and diversity, i.e., NCircles.

## 4 EXPERIMENT

In this section, we validate our supervised fine-tuning and reinforcement learning methods for generating diverse molecules, coined Div-SFT and Div-SFT+RL, respectively. In these experiments, one can observe that (1) our fine-tuning approach enables LLMs to better discover diverse molecules compared to decoding schemes for diversified generation and (2) our fine-tuned LLM outperforms other representative LLMs, including generalist and specialist models for chemical tasks.

### 4.1 COMPARISON WITH DECODING SCHEMES

In this experiment, we show that our method enables LLMs to better generate diverse molecules compared to the existing decoding schemes for diverse sequence generation. To validate the consistent improvement, we implement our fine-tuning method and decoding schemes on two models specialized in the chemical domain: BioT5$^+$ (Pei et al., 2024) and MolT5 (Edwards et al., 2022). In measuring metrics, we apply canonicalization to the SMILES representations to remove the randomness stemming from the non-uniqueness of SMILES (Weininger et al., 1989).

**Tasks.** We consider description-guided molecule generation using the ChEBI-20 dataset, which includes training and test sets (Edwards et al., 2021). Note that the training dataset has also been used to pre-train the base LLMs, BioT5$^+$ and MolT5, i.e., not an external dataset. The test dataset is unobserved during both the pre-training and fine-tuning phases. Each data point consists of a pair of a description and an example of target molecule that satisfies the description.

In our experiments, we consider generating 50 molecules for each description in the ChEBI-20 test dataset involving 3300 molecular descriptions. We then evaluate both the structural diversity of the generated molecules and how well they satisfy the given descriptions.

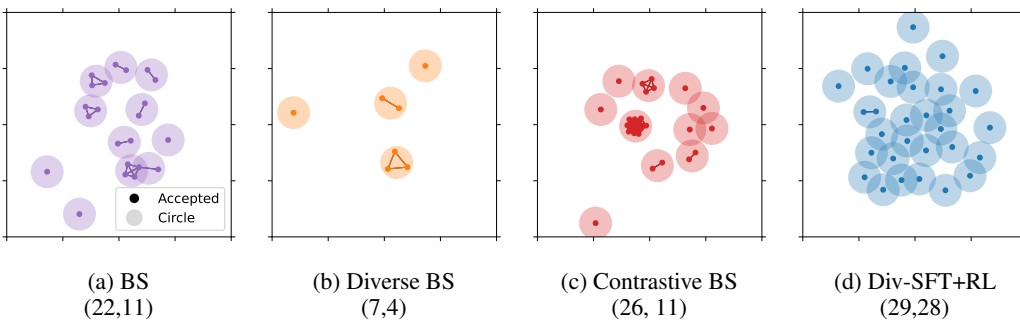

| (a) BS | (b) Diverse BS | (c) Contrastive BS | (d) Div-SFT+RL |
|:------:|:--------------:|:------------------:|:--------------:|
| (22,11) | (7,4) | (26, 11) | (29,28) |

Figure 6: **The distribution of (Accepted & unique, NCircles$_{h=0.65}$) obtained by each method.** The molecules are generated from the description in Table 15. The Fruchterman-Reingold force-directed algorithm (Fruchterman & Reingold, 1991) embeds each molecule in a 2-dimensional space based on Tanimoto similarity. Here, a low distance yields a high Tanimoto similarity. Each dot, shaded circle, and edge represent accepted molecules, a boundary for determining similar or distinct molecules, and a pair with similarity above 0.65, respectively. Our method better captures the molecular diversity.

**Metrics.** To measure the similarity between two molecules, we use Tanimoto similarity (Bajusz et al., 2015) which measures the distance between molecular structural features, e.g., the presence of specific substructures. The detailed explanation is described in Appendix A.2. Based on this, we consider the following metrics for generated molecules:

- The number of accepted and unique molecules (Accepted & unique): This metric measures the number of valid and unique molecules satisfying the given description. Following the metric of prior studies (Edwards et al., 2022; Pei et al., 2024), we evaluate the molecule as satisfying the given description when its BLEU score between an example of target molecule is high ($> 0.7$) [6]

- The number of circles (NCircles; Xie et al., 2023): This metric considers both quality and diversity. Given the set of accepted molecules, the NCircles$_h$ computes the size of the largest subset in which no two molecules are similar to each other (Tanimoto similarity below a threshold $h$), i.e., this measures the volume of chemical space covered by a given set. The detailed description follows Appendix A.3. We also illustrate Figure 6 for explaining NCircles in a 2-dimensional space.

- Internal diversity (IntDiv.; Polykovskiy et al., 2020): This metric is the complement of the average of pair-wise Tanimoto similarities between the accepted molecules satisfying the given description. The detailed description follows Appendix A.3.

- Average of top 10 scores (Top 10): This metric measures the quality of the generated molecules by averaging the BLEU scores (Papineni et al., 2002) of the top unique 10 molecules yielding high BLEU scores between the ground-truth example.

**Implementations.** For supervised fine-tuning, we collect a hundred molecules for each molecular description $p_{\text{desc}}$ in ChEBI-20 training sets. Then, they are filtered to remove invalid molecules, duplicated molecules, and unaccepted molecules. The description-matching reward $r_{\text{match}}(m_k)$ is defined as a BLEU score between a ground-truth example. The diversity reward $r_{\text{div}}(m_k, \{m_i\}_{i=1}^{k-1})$ is defined as the complement of the maximum Tanimoto similarity between generated molecules $\{m_i\}_{i=1}^{k-1}$. The detailed implementations and hyper-parameters follow Appendix B.

**Baselines.** We compare our fine-tuning approach with various decoding schemes. We consider the random sampling with different temperatures $\{0.7, 1.0, 1.5\}$ and nucleus sampling (Holtzman et al., 2020). We also consider beam search (BS) and the variants of BS to promote sequence-level diversity between the generated samples: diverse BS (Vijayakumar et al., 2018) and contrastive BS (Su et al., 2022). The detailed settings are described in Appendix C.

**Results.** We present the main results on BioT5$^+$ and MolT5 in Figure 5(a) and Figure 5(b), respectively. One can see that our approach, i.e., Div-SFT+RL, shows superior performance compared to the considered baselines. Especially, it is worth noting that our approach makes significant improvements in NCircles metrics that require generating diverse molecules satisfying the given description.

---

[6]However, the BLEU score has a limitation by just measuring the textual similarity. In Appendix D.2, we discuss this and conduct experiments by replacing BLEU with Tanimoto and Dice scores.

Table 1: **Visualization of the generated molecules with their diversity.** The eight molecules are generated with the description: "The molecule is a primary aliphatic ammonium ion which is obtained from streptothricin F by protonation of the guanidino and amino groups. It has a role as an antimicrobial agent. It is a guanidinium ion and a primary aliphatic ammonium ion. It is a conjugate acid of a streptothricin F'." The generated molecule with blue line indicates the accepted molecule.

| Method (IntDiv.) | Example of outputs |
|---|---|
| BS (0.51) |  |
| Diverse BS (0.28) |  |
| Contrastive BS (0.53) |  |
| Div-SFT+RL (0.69) |  |

Table 2: **Comparison with existing LLMs on description-based molecule generation.** Our fine-tuned model shows superior performance compared to the considered baselines.

| Method | Accepted & Unique | NCircles$_{h=0.75}$ | NCircles$_{h=0.65}$ | IntDiv. | Top 10 |
|---|---|---|---|---|---|
| *Chemical-task Specialist LLMs* | | | | | |
| MolT5 | 3.12 | 1.85 | 1.59 | 0.29 | 0.41 |
| Text+Chem T5 | 13.78 | 4.82 | 3.29 | 0.23 | 0.63 |
| BioT5 | 15.58 | 8.45 | 6.07 | 0.33 | 0.69 |
| BioT5$^+$ | 16.03 | 7.93 | 6.16 | 0.31 | 0.63 |
| *Generalist LLMs* | | | | | |
| Mol-instructions | 0.06 | 0.03 | 0.03 | 0.02 | 0.01 |
| LlaSMol | 17.68 | 6.73 | 4.94 | 0.26 | 0.68 |
| GPT-3.5-turbo | 3.87 | 2.83 | 2.49 | 0.20 | 0.41 |
| GPT-4o | 5.74 | 4.09 | 3.53 | 0.22 | 0.48 |
| o1-preview | 6.92 | 3.46 | 2.36 | 0.13 | 0.31 |
| **BioT5$^+$+ours** | **21.98** | **16.98** | **14.35** | **0.45** | **0.74** |

We also provide the qualitative results in Figure 6 and Table 15 with sampled molecules. One can see that variants of beam search for diverse sentence generation, i.e., diverse BS and contrastive BS, do not enhance molecular structural diversity. In contrast, our approach demonstrates significantly higher molecular structural diversity. In Appendix D, we provide additional examples of molecules generated by the considered baselines and our method.

## 4.2 COMPARISON WITH EXISTING LLMS

Here, we compare our fine-tuned BioT5$^+$, presented in Figure 5, with various recent LLMs, including chemical-task specialists, generalists, and fine-tuned generalist LLMs on chemical domains. The purpose of this experiment is to highlight the limitations of existing LLMs in capturing molecular diversity, whereas our fine-tuned model successfully captures molecular diversity during generation. The tasks and metrics are the same as settings in Section 4.1. We use the first 500 molecular descriptions in the ChEBI-20 test dataset for evaluation.

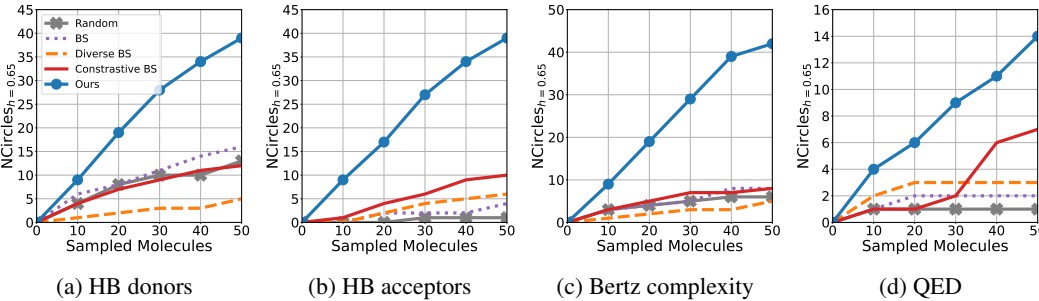

| (a) HB donors | (b) HB acceptors | (c) Bertz complexity | (d) QED |

Figure 7: **Experiments on DrugAssist.** Our method consistently improves the performance for generating diverse and high-quality molecules when implemented on generalist LLMs.

**Baselines.** We compare our fine-tuned model with various existing LLMs. We consider four LLMs specialized for chemical tasks: Text+Chem T5 (Christofidellis et al., 2023), MolT5, BioT5 (Pei et al., 2023), and BioT5$^+$. Next, we consider two generalist LLMs: Mol-instructions (Fang et al., 2024) fine-tuned from Llama-7B (Touvron et al., 2023), and LlaSMol (Yu et al., 2024) fine-tuned from Mistral-7B (AI, 2023), based on description-based molecular generation tasks. For each baseline, we report the best results (highest NCircles$_{h=0.65}$) obtained using either random sampling, beam search, diverse beam search, or contrastive beam search. We also consider three strong API-based generalist LLMs: GPT-3.5, GPT-4o, and o1-preview (OpenAI, 2023; 2024).[7] The detailed experimental settings and prompts are described in Appendix C.

**Results.** We present the results in Table 2. One can see that our fine-tuned model shows superior performance compared to the considered baselines in discovering diverse and high-quality molecules. Furthermore, it is noteworthy that most existing LLMs yield low NCircles with respect to the number of generated accepted and unique molecules, while our method yields relatively high NCircles.

### 4.3 Fine-tuning generalist LLMs

We further validate whether our fine-tuning method improves the generalist LLMs as well in terms of the diversity of generated molecules. Here, as a base LLM for implementing our method, we consider DrugAssist (Ye et al., 2023) which is fine-tuned from the Llama-7B. As baselines, we apply random sampling and contrastive beam search to DrugAssist.

**Tasks.** In this experiment, we consider prompts based on the four quantitative molecule properties: hydrogen bond (HB) donors, HB acceptors, Bertz complexity (Bertz, 1981), and quantitative estimate of drug-likeness (QED) (Bickerton et al., 2012). The goal of this task is to generate diverse molecules that satisfy the properties described in the given prompt.

**Implementations.** We apply our supervised fine-tuning and reinforcement learning to enhance DrugAssist. For supervised fine-tuning, we collect multiple molecules for prompts about three properties: HB donors, HB acceptors, and Bertz complexity. The prompts about QED are excluded from training but included in the evaluation to assess generalization to unseen properties. The sampled molecules are filtered to remove invalid molecules, duplicated molecules, and molecules that do not satisfy the given properties. The property-match reward $r_{\text{match}}(m_k)$ yields a non-zero value when satisfying the given property, and the diversity reward $r_{\text{div}}(m_k, \{m_i\}_{i=1}^{k-1})$ is defined as same as Section 4.1. The detailed implementations and hyper-parameters follow Appendix B.

**Results.** We present the results in Figure 7. One can see that our approach consistently improves the performance in generating diverse and high-quality molecules when implemented on generalist LLMs. Note that our approach consistently demonstrates superior performance for the unseen prompt, i.e., the prompt about QED in Figure 7(d).

### 4.4 Ablation studies

**Large number of samples vs. performance.** We also analyze how well our method discovers diverse molecules with respect to the number of generations. In this experiment, we extend beyond

---

[7]We use `gpt-3.5-turbo-0125` and `gpt-4o-2024-08-06`.

Table 3: **Experiments with the large number of samples.** The base LLM is BioT5$^+$. Our method discovers more diverse molecules with respect to the (1) the number of generations and (2) time costs.

| Method$_{\text{num. of generations}}$ | BS$_{300}$ | BS$_{400}$ | BS$_{500}$ | Ours$_{85}$ | Ours$_{120}$ | Ours$_{155}$ |
|---|---|---|---|---|---|---|
| NCircles$_{h=0.65}$ | 18.6 | 20.4 | 21.3 | 20.4 | 23.6 | 25.6 |
| Time (sec.) | 323 | 452 | 585 | 65 | 86 | 107 |

Table 4: **Comparison with variants of implementations.** The base LLM is BioT5$^+$. Applying multi-stage reinforcement learning shows superior performance compared to (1) supervised fine-tuning with hard filtering and (2) single-stage reinforcement learning.

| Method | Accepted & Unique | NCircles$_{h=0.75}$ | NCircles$_{h=0.65}$ | IntDiv. | Top 10 |
|---|---|---|---|---|---|
| Div-SFT$_{\text{hard}}$ | 9.23 | 7.46 | 6.15 | 0.33 | 0.59 |
| Div-SFT+RL$_{\text{single}}$ | 14.16 | 8.50 | 6.49 | 0.29 | 0.68 |
| Div-SFT+RL | 21.98 | 16.98 | 14.35 | 0.45 | 0.74 |

the settings in Section 4.1. We consider our method to generate 85, 120, and 155 molecules with a single NVIDIA A100 SXM4 40GB GPU. Next, we consider beam search with beam sizes of 300, 400, and 500, respectively. The beam search is implemented with four NVIDIA A100 SXM4 40GB GPUs due to the memory limitation of a single GPU. Note that this experiment uses 250 molecular descriptions in the ChEBI-20 test set. We present the results in Table 3. One can see that our method (1) discovers more diverse molecules with respect to the total number of generations and (2) exhibits further performance improvements as the number of generations increases.

**Time costs vs. performance.** In addition, we also analyze how well our method discovers diverse molecules with respect to the time costs. In Table 3, we present the time costs for each method. One can see that our method discovers more diverse molecules in a practical time compared to the decoding schemes for diverse sequence generation.

**SFT with hard filtering vs. RL.** To train LLMs to generate diverse molecules, one may also consider supervised fine-tuning on hard-filtered datasets, e.g., using a set of molecules filtered by similarity, as an alternative to reinforcement learning. In this experiment, we additionally perform supervised fine-tuning with distinct molecules, where each pair of molecules yields a Tanimoto similarity below 0.65. The results are presented in Table 4. One can see that while it yields relatively high NCircles with respect to the number of accepted unique molecules, the overall performance is worse compared to the performance of reinforcement learning.

**Single-stage vs. multi-stage RL.** As mentioned in Section 3.2, we consider a multi-stage setting for generating multiple molecules. However, one may also consider a single-stage setting, where the return of a generated sequence is defined as the sum of the rewards from multiple generated molecules. In Table 4, we compare both approaches. One can see that the multi-stage setting significantly outperforms the single-stage setting. We hypothesize that this result stems from credit assignment issues in the single-stage setting. Namely, the single-stage setting lacks signals to capture molecule-wise impacts on diversity among a large set of molecules and fails to promote the generation of molecules responsible for increasing diversity.

## 5 CONCLUSION

In this paper, we identify the limitations of large language models (LLMs) for generating diverse molecules. Then, we present new supervised fine-tuning and reinforcement learning methods to adapt existing molecular generative LLMs to generate a diverse set of molecules. Experiments show that our approach enables LLMs to better discover diverse molecules compared to the existing approaches. An interesting avenue for future work is to develop a concrete benchmark for text-guided diverse molecular generation, as current ChEBI-20 datasets were originally designed for a single molecular generation. Another interesting avenue is to reduce the space and time complexity in generating the sequence of molecules, for example, based on set encoding techniques (Zaheer et al., 2017).

**Reproducibility.** We describe experimental details in Appendices B to C that include detailed hyper-parameters and prompts. In the supplementary materials, we provide the code for fine-tuning BioT5$^+$, which involves both supervised fine-tuning and reinforcement learning.

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

# A  ADDITIONAL RELATED WORKS

## A.1  REINFORCEMENT LEARNING (RL) FOR DIVERSITY IN MOLECULAR GENERATION

Existing literature has studied RL-based methods to generate molecules with desired properties while enhancing their diversity. First, Blaschke et al. (2020); Pereira et al. (2021) introduced memory-assisted RL, which penalizes the reward of a molecule when it is highly similar to the molecules stored in the memory unit. He et al. (2024) also incorporated RL with a diversity penalty in transformer-based architectures for molecular generation. In addition, Hu et al. (2024) leveraged multiple GPT-based agents trained with RL to encourage these agents to explore diverse directions for discovering diverse molecules. Their algorithms are designed to discover diverse molecules with a fixed target property. In contrast, our work fine-tunes LLMs to generate diverse molecules given a prompt that is flexible to describe various target properties.

## A.2  MOLECULAR SIMILARITY MEASURES

In this section, we explain the measures for evaluating the similarity between two molecules. These measures are used to define the reward of reinforcement learning (Appendix B) and diversity metrics (Appendix A.3). Specifically, the molecular similarity is evaluated with their Morgan fingerprint (Rogers & Hahn, 2010), which is a vector characterizing the presence of specific atoms, bonds, or substructures. Then, the similarity between two molecules is typically evaluated as follows::

$$T(m_i, m_j) = \frac{|f(m_i) \cap f(m_i)|}{|f(m_i) \cup f(m_j)|},$$

where $f(m_i)$ maps the molecule $m_i$ to its Morgan fingerprint. This similarity $T(m_i, m_j)$ is referred to as the Tanimoto similarity between $m_i$ and $m_j$, which focuses on evaluating structural similarity.

**Other similarity measures.** To evaluate the molecular similarity, one can use other measures for computing the similarities between two fingerprints. For example, Dice and cosine similarities (Bajusz et al., 2015) are defined as follows:

$$D(m_i, m_j) = \frac{2|f(m_i) \cap f(m_i)|}{|f(m_i)| + |f(m_j)|}, \quad C(m_i, m_j) = \frac{\langle f(m_i), f(m_j) \rangle}{|f(m_i)||f(m_j)|},$$

where $\langle \cdot, \cdot \rangle$ denotes a dot product between two vectors.

## A.3  MOLECULAR DIVERSITY METRICS

In this section, we provide a detailed explanation of the diversity metrics for evaluating the given set of molecules. Specifically, we explain two diversity metrics: the number of circles (Xie et al., 2023) and the internal diversity (Polykovskiy et al., 2020).

**The number of circles (NCircles.; Xie et al., 2023).** To evaluate the diversity of a given set of molecules $\mathcal{M}$, this computes the size of the largest subset of molecules in which no two molecules are similar to each other. Specifically, this metric is defined with a Tanimoto similarity $T(\cdot, \cdot)$ and a similarity threshold $h$ as follows:

$$\text{NCircles}_h = \max_{\mathcal{C} \subseteq \mathcal{M}} |\mathcal{C}| \quad \text{s.t.} \ T(x, y) < h, \forall x \neq y \in \mathcal{C}, \tag{2}$$

where $\mathcal{C}$ is a subset of molecules. Every pair of molecules in $\mathcal{C}$ should have a similarity lower than $h$. The high NCircles value implies that the given set of molecules $\mathcal{M}$ is diverse and covers a wide range of molecular space.

**Internal diversity (IntDiv.; Polykovskiy et al., 2020).** Given a set of molecules $\mathcal{M}$, this metric measures the average of pair-wise Tanimoto similarities to evaluate the overall diversity. Specifically, the IntDiv. is defined as follows:

$$\text{Intdiv.} = \frac{1}{|\mathcal{M}| \cdot (|\mathcal{M}| - 1)} \sum_{i=1}^{|\mathcal{M}|} \sum_{j=i+1}^{|\mathcal{M}|} (1 - T(m_i, m_j)), \tag{3}$$

where $m_i$ is $i$-th molecule in the given set of molecules $\mathcal{M}$.

## B DETAILED IMPLEMENTATIONS AND TRAINING

### B.1 SUPERVISED FINE-TUNING

**Dataset Collection.** To fine-tune BioT5$^+$ and MolT5 (Sections 4.1 and 4.2), we collect $T = 100$ molecules using beam search for each training molecular description in the ChEBI-20 training dataset. Note that this dataset has been considered in the original BioT5$^+$ and MolT5.

For the fine-tuning of DrugAssist (Section 4.3), we collect $T = 300$ molecules using beam search for each training prompt. Note that the collected molecules were filtered to remove invalid string representations, duplicated molecules, and molecules that do not satisfy the given description. The invalid string representations are evaluated with RDKit package (Landrum, 2016). Additionally, the collected molecules are concatenated into a single sequence $m_1 || \cdots || m_K$. Note that two molecules are separated by introducing a newline character ('\n').

**Supervised learning.** We consider four NVIDIA A100 GPUs for supervised fine-tuning.

- For the supervised fine-tuning of BioT5$^+$ and MolT5 (Sections 4.1 and 4.2), we consider 80 epochs, 8-batch size, $5e - 4$ learning rate, $0.05$ warm-up ratio, and apply a cosine learning scheduler. The maximum sequence length in supervised training is limited to 2560 due to memory limitations.

- For the supervised fine-tuning of DrugAssist (Section 4.3), we consider 80 epochs, 4-batch size, $3e - 5$ learning rate, $0.05$ warm-up ratio, and apply a cosine learning scheduler. The maximum sequence length in supervised training is limited to 1024 due to memory limitations. We also apply LoRA (Hu et al., 2022), where the rank and alpha are 64 and 128, respectively.

### B.2 REINFORCEMENT LEARNING

**Reward Design.** In experiments with BioT5$^+$ and MolT5 on the ChEBI-20 dataset (Sections 4.1 and 4.2), we define the description-matching reward using the BLEU score as follows:

$$r_{\text{match}}(m_k, p_{\text{desc}}) = \text{BLEU}(m_{p_{\text{mol}}}, m_k)^\alpha, \tag{4}$$

where $m_{p_{\text{mol}}}$ is a ground-truth molecule satisfying the given description. The ChEBI-20 dataset involves a set of pairs $(p_{\text{desc}}, m_{p_{\text{mol}}})$. Note that $\alpha$ is a hyper-parameter.

For experiments with DrugAssist (Section 4.3), the property-matching reward yields 1 if the molecule satisfies the quantitative properties described in $p_{\text{desc}}$, e.g., HB donors, HB acceptors, and Bertz complexity, and 0 otherwise. The properties are evaluated with the RDKit package (Landrum, 2016).

Next, the diversity reward, $r_{\text{div}}$, is defined to consider molecular structural diversity as follows:

$$r_{\text{div}}(m_k, \{m_i\}_{i=1}^{k-1}) = 1 - \max_{m \in \{m_i\}_{i=1}^{k-1}} T(m_k, m)^\beta, \tag{5}$$

where $T(m_i, m_j)$ measures the Tanimoto similarity between $m_i$ and $m_j$ by assessing their true molecular structures, i.e., fingerprints. This metric yields a value between 0 and 1 and can be obtained using the RDKit package. Note that $\beta$ is a hyper-parameter.

**Policy optimization.** We consider four NVIDIA A100 SXM4 40GB for reinforcement learning implemented with proximal policy optimization.

- For the reinforcement learning of BioT5$^+$ and MolT5 (Sections 4.1 and 4.2), we consider 200 PPO iterations, 8 mini-batch size, 128 batch size, and $5e - 5$ learning rate. We also consider $0.01$ KL penalty. Note that $\alpha$ and $\beta$ in Equations (4) and (5) are 0.5 and 2.0, respectively. The reward signal is amplified by multiplying by a value of $8.0$. The maximum sequence length in reinforcement learning is limited to 2560 due to memory limitations. Here, we also apply LoRA (Hu et al., 2022) where the rank and alpha are 16 and 32, respectively. We save the model every 25 PPO iteration and select the model yielding the highest rewards for the training prompts.

- For the reinforcement learning of DrugAssist (Section 4.3), we consider 200 PPO iterations, 4 mini-batch size, 64 batch size, and $3e - 6$ learning rate. We also consider $0.1$ KL penalty. Note that $\beta$ in Equation (5) is 2.0. The reward signal is amplified by multiplying by a value of $4.0$. The maximum sequence length in reinforcement learning is limited to 1280. We also apply LoRA (Hu et al., 2022) where the rank and alpha are 64 and 128, respectively. We save the model every 25 PPO iteration and select the model yielding the highest rewards for the training prompts.

## C  DETAILED EXPERIMENTAL SETTINGS

**Comparison with decoding schemes (Section 4.1).** In this experiment, we first consider random sampling with different temperatures $\{0.7, 1.0, 1.5\}$. For the other decoding schemes, we consider conventional configurations: nucleus sampling with top-p $0.8$, beam search, diverse beam search with a diversity penalty of $0.5$, and contrastive beam search with a penalty alpha of $0.5$. We apply greedy decoding for our approach. The prompts for BioT5$^+$ are described in Table 5. The prompts for MolT5 are considered as a molecular description without any additional comments.

Table 5: **Prompts for BioT5$^+$ (Pei et al., 2024).**

| Prompt | Contents |
|---|---|
| $p_{\text{desc}}$ | "Definition: You are given a molecule description in English. Your job is to generate the molecule SELFIES that fits the description.
Now complete the following example -
Input: \<molecular description\>
Output: " |
| $p_{\text{desc+div}}$ (fine-tuning) | "Definition: You are given a molecule description in English. Your job is to generate the molecule SELFIES that fits the description.
Now provide a set of molecules -
Input: \<molecular description\>
Output: " |

**Comparison with LLMs (Section 4.2).** In this experiment, we consider generalist LLMs. For Mol-instructions, and LlasMol, we apply random sampling, beam search, diverse beam search with a diversity penalty of $0.5$, and contrastive beam search with a penalty alpha of $0.5$. The prompts are described in Table 6. For ChatGPT, we apply either $50$ random sampling with $p_{\text{desc}}$ or greedy decoding with $p_{\text{desc+div}}$, where both prompts are described in Table 7. The prompt design is based on (Li et al., 2024). For o1, we use the prompt $p_{\text{desc+div}}$ described in Table 7. We report the test score using the decoding scheme or the prompt that yields the highest NCircles$_{h=0.65}$ for $500$ evaluation molecular description.

Table 6: **Prompts for generalist LLMs (Fang et al., 2024; Taylor et al., 2022).**

| Method | $p_{\text{desc}}$ |
|---|---|
| Mol-instructions | "Below is an instruction that describes a task, paired with an input that provides further context. Write a response that appropriately completes the request.
### Instruction:
Create a molecule with the structure as the one describe.
### Input:
\<molecular description\>
### Response: " |
| LlasMol | "Give me a molecule that satisfies the conditions outlined in the description: \<molecular description\>" |

Table 7: **Prompts for ChatGPT and o1 (Li et al., 2024).**

| Input | Contents |
|---|---|
| System $p_{desc}$ | "You are now working as an excellent expert in chemistry and drug discovery. Given the caption of a molecule, your job is to predict the SMILES representation of the molecule. The molecule caption is a sentence that describes the molecule, which mainly describes the molecule's structures, properties, and production. You can generate the molecule SMILES representation from the caption. Example: Instruction: Given the caption of a molecule, predict the SMILES representation of the molecule. Input: <molecular description>. Your output should be: {"molecule": "<SMILES>"} Your response should only be in the JSON format above; THERE SHOULD BE NO OTHER CONTENT INCLUDED IN YOUR RESPONSE." |
| User $p_{desc}$ | "Input: <molecular description>" |
| System $p_{desc+div}$ | "You are now working as an excellent expert in chemistry and drug discovery. Given the caption of a molecule, your job is to generate diverse molecules. The molecule caption is a sentence that describes the molecule, which mainly describes the molecule's structures, properties, and production. You can generate the molecule SMILES representation from the caption. Example: Instruction: Given the caption of a molecule, generate fifty structurally diverse molecules with SMILES representations. Input: <molecular description>. Your output should be: {"molecule1": "<SMILES>"} {"molecule2": "<SMILES>"} {"molecule3": "<SMILES>"} ... {"molecule50": "<SMILES>"} Your response should only be in the JSON format above; THERE SHOULD BE NO OTHER CONTENT INCLUDED IN YOUR RESPONSE." |
| User $p_{desc+div}$ | "Input: <molecular description>" |

**Fine-tuning generalist LLMs.** We synthesize 600 pairs of training prompts and corresponding sets of molecules. The prompts specify hydrogen bond donors and acceptors ranging from one to four, and a Bertz complexity ranging from 0 to 300. The sets of molecules are collected by applying beam search on DrugAssist and then perturbed by shuffling their order. We use the prompts described in Table 8. The system prompt follows the default settings from (Ye et al., 2023). For evaluation, we use four prompts specifying three hydrogen bond donors, three hydrogen bond acceptors, a Bertz complexity between 100 and 200, and a QED value between 0.4 and 0.6. Additionally, as shown in Figure 3, we try to generate diverse molecules by designing prompts without fine-tuning (Table 9). However, these prompts show lower performance compared to applying beam search.

Table 8: **Prompts for DrugAssist (Ye et al., 2023).**

| Prompt | Contents |
|---|---|
| $p_{desc}$ | Hydrogen bond donors and acceptors: "Can you generate a molecule with \<value\> \<property\>? Print it in SMILES format." QED and Bertz complexity: "Can you generate a molecule with \<property\> below \<value1\> but at least \<value2\>? Print it in SMILES format." |
| $p_{desc+div}$ (fine-tuning) | Hydrogen bond donors and acceptors: "Can you generate a set of molecules that have \<value\> \<property\>? Print each of them in SMILES format." QED and Bertz complexity: "Can you generate a set of molecules that have \<property\> below \<value1\> but at least \<value2\>? Print each of them in SMILES format." |

Table 9: **Prompts for DrugAssist (without fine-tuning, Figure 3).**

| $p_{desc+div}$ |
|---|
| "Can you generate a set of molecules? Each molecule has \<value\> \<property\>. Print each of them in SMILES format." |
| "Can you generate a diverse set of molecules? Each molecule has \<value\> \<property\>. Print each of them in SMILES format." |
| "Can you generate a structurally diverse set of molecules? Each molecule has \<value\> \<property\>. Print each of them in SMILES format." |
| "Can you generate fifty molecules? Each molecule has \<value\> \<property\>. Print each of them in SMILES format." |
| "Can you generate fifty diverse molecules? Each molecule has \<value\> \<property\>. Print each of them in SMILES format." |
| "Can you generate fifty structurally diverse molecules? Each molecule has \<value\> \<property\>. Print each of them in SMILES format." |
| "Can you generate a set of molecules with \<value\> \<property\>? Print each of them in SMILES format." |
| "Can you generate a diverse set of molecules with \<value\> \<property\>? Print each of them in SMILES format." |
| "Can you generate a structurally diverse set of molecules with \<value\> \<property\>? Print each of them in SMILES format." |
| "Can you generate fifty molecules with \<value\> \<property\>? Print each of them in SMILES format." |
| "Can you generate fifty diverse molecules with \<value\> \<property\>? Print each of them in SMILES format." |
| "Can you generate fifty structurally diverse molecules with \<value\> \<property\>? Print each of them in SMILES format." |

# D  ADDITIONAL RESULTS

## D.1  EXPERIMENTS WITH THE LARGE NUMBER OF GENERATION

Table 10: **Experiments with the large number of samples.** The base LLM is BioT5$^+$. Our method discovers more diverse molecules with respect to the (1) the number of generations and (2) time costs.

| Method$_{\text{num. of generations}}$ | BS$_{300}$ | BS$_{400}$ | BS$_{500}$ | Ours$_{85}$ | Ours$_{120}$ | Ours$_{155}$ |
|---|---|---|---|---|---|---|
| NCircles$_{h=0.75}$ | 25.84 | 28.15 | 31.41 | 26.45 | 32.13 | 37.38 |
| NCircles$_{h=0.65}$ | 18.6 | 20.4 | 21.3 | 20.4 | 23.6 | 25.6 |
| Accepted & Unique | 54.65 | 62.69 | 69.19 | 34.20 | 41.44 | 48.20 |
| Top 10 | 0.69 | 0.69 | 0.70 | 0.76 | 0.77 | 0.78 |
| Intdiv. | 0.33 | 0.33 | 0.32 | 0.47 | 0.48 | 0.49 |
| Time (sec.) | 323 | 452 | 585 | 65 | 86 | 107 |

In Table 10, we provide full results of Table 3. One can see that our method shows superior performance in the NCircles metrics which capture both quality and diversity, the average of the top 10 scores, and internal diversity with respect to the number of generations and time costs.

## D.2  EXPERIMENTS WITH ACCEPTANCE BASED ON DICE SIMILARITY

Table 11: **Experiments with acceptance based on Tanimoto or Dice similarities.** The base LLM is BioT5$^+$. Our method is superior in discovering diverse molecules.

| Method | Tanimoto sim. | | Dice sim. | |
|---|---|---|---|---|
| | Accepted & Unique | NCircles$_{h=0.75}$ | Accepted & Unique | NCircles$_{h=0.75}$ |
| Random | 1.7 | 1.1 | 2.0 | 1.4 |
| BS | 9.3 | 2.9 | 11.7 | 4.2 |
| Diverse BS | 4.7 | 2.6 | 5.9 | 3.6 |
| Div-SFT+RL | 10.0 | 5.8 | 12.6 | 8.0 |

In Sections 4.1 and 4.2, we evaluate whether the molecule satisfies the given description by measuring the BLEU score between the generated molecule and the target molecule. However, the BLEU score has a limitation: just measures the textual similarity of SMILES strings and may fail to capture molecular structural or functional similarities with the target molecule. To address this, in Table 11, we also conduct experiments by replacing the BLEU score in the metrics with Tanimoto similarity and Dice similarity, which are more concrete metrics for capturing molecular structure and function.[8] Notably, our method, trained with BLEU scores as rewards, still outperforms the baselines.

---

[8]We consider a molecule accepted if the Tanimoto and Dice similarities between the target molecule are higher than 0.6 and 0.7, respectively.

Table 12: **Visualization of the generated molecules from beam search.** The 48 molecules are generated with the description: "The molecule is a primary aliphatic ammonium ion which is obtained from streptothricin F by protonation of the guanidino and amino groups. It has a role as an antimicrobial agent. It is a guanidinium ion and a primary aliphatic ammonium ion. It is a conjugate acid of a streptothricin F'."

| Example of outputs (IntDiv. is 0.53) |
| --- |

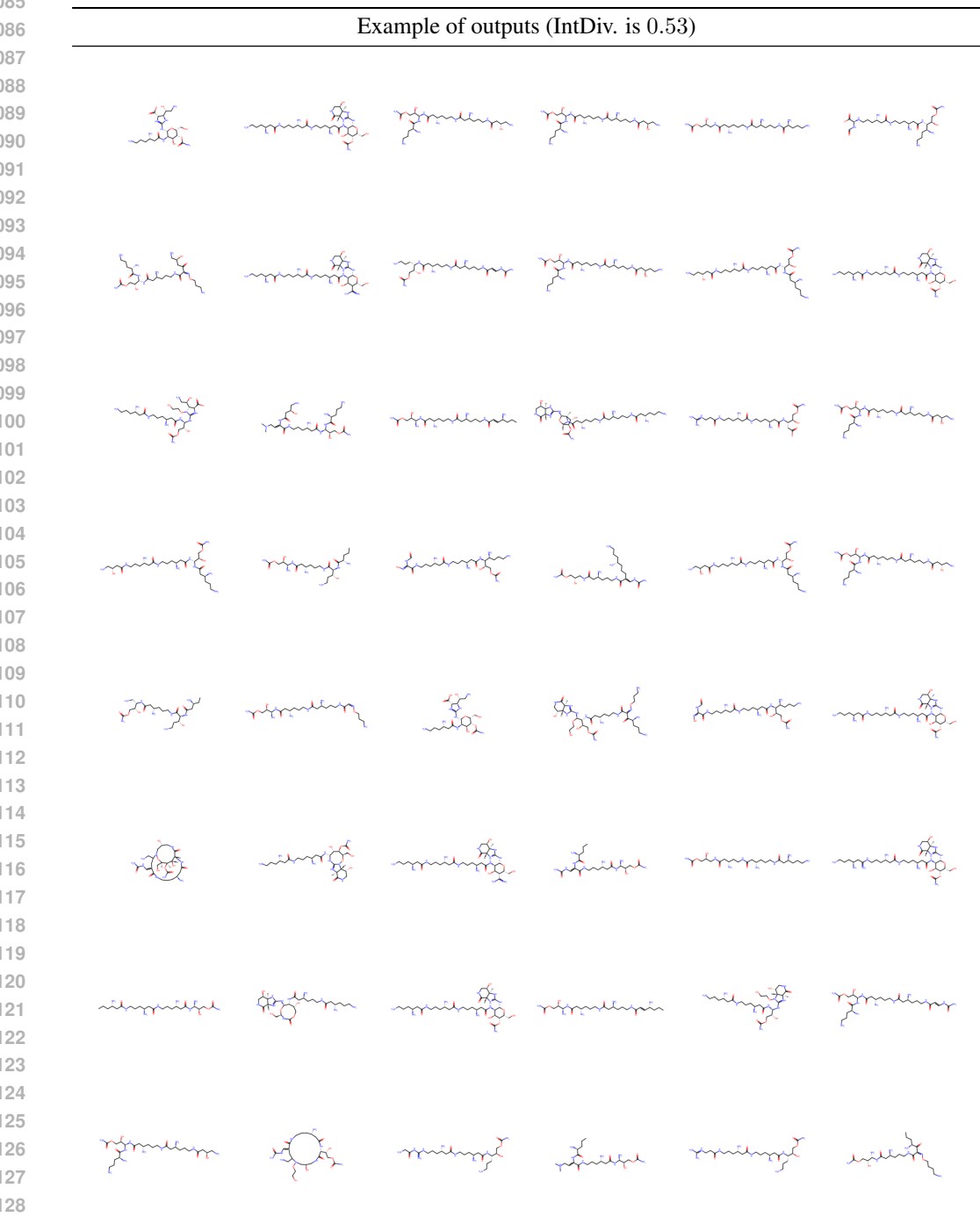

Table 13: **Visualization of the generated molecules from diverse beam search.** The 48 molecules are generated with the description: "The molecule is a primary aliphatic ammonium ion which is obtained from streptothricin F by protonation of the guanidino and amino groups. It has a role as an antimicrobial agent. It is a guanidinium ion and a primary aliphatic ammonium ion. It is a conjugate acid of a streptothricin F'."

| Example of outputs (IntDiv. is 0.67) |
| --- |

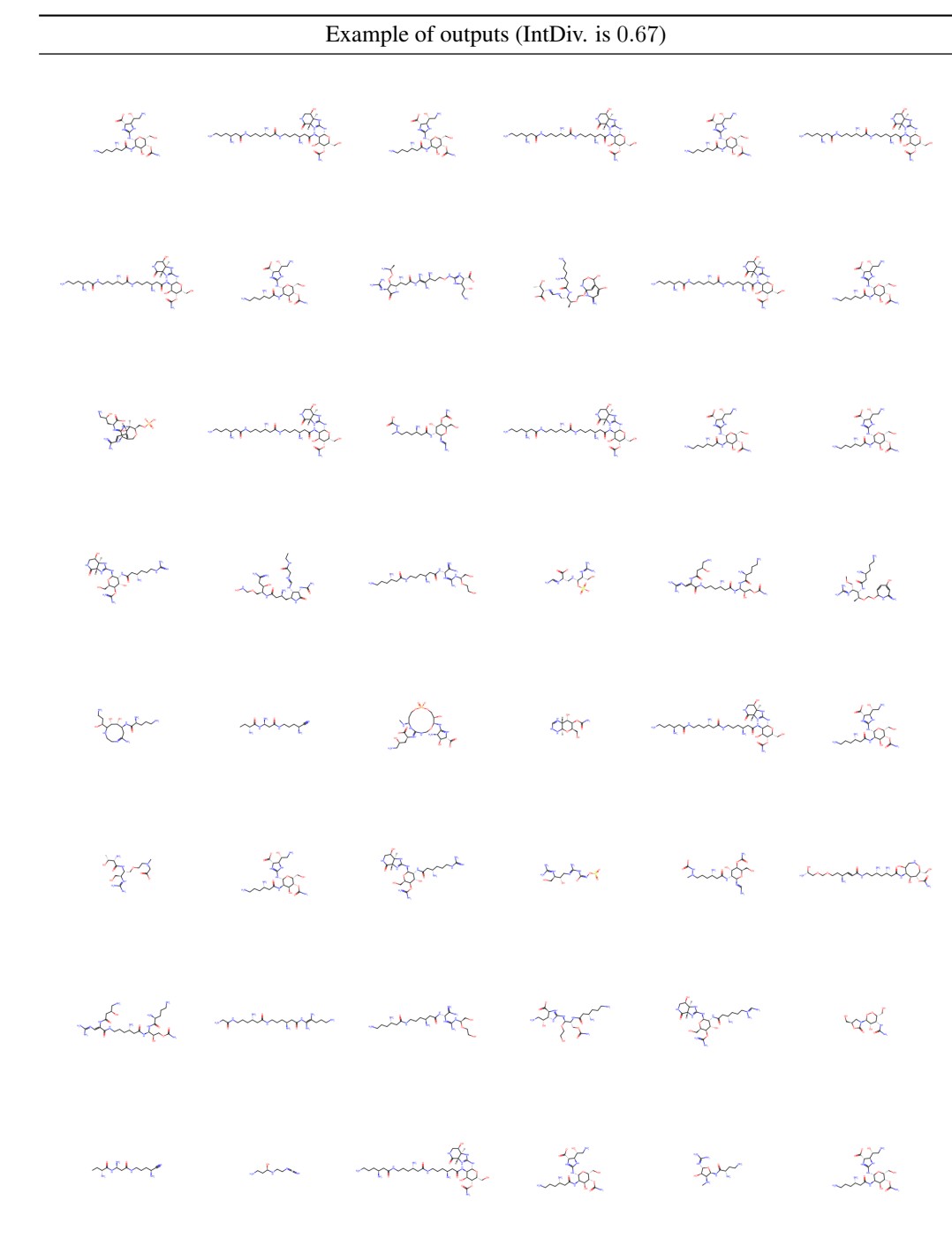

Table 14: **Visualization of the generated molecules from constrastive beam search.** The 48 molecules are generated with the description: "The molecule is a primary aliphatic ammonium ion which is obtained from streptothricin F by protonation of the guanidino and amino groups. It has a role as an antimicrobial agent. It is a guanidinium ion and a primary aliphatic ammonium ion. It is a conjugate acid of a streptothricin F'."

| Example of outputs (IntDiv. is 0.58) |
| --- |

Table 15: **Visualization of the generated molecules from our method.** The 48 molecules are generated with the description: "The molecule is a primary aliphatic ammonium ion which is obtained from streptothricin F by protonation of the guanidino and amino groups. It has a role as an antimicrobial agent. It is a guanidinium ion and a primary aliphatic ammonium ion. It is a conjugate acid of a streptothricin F."

| Example of outputs (IntDiv. is 0.77) |
| --- |

