# OpenReview forum: "Can LLMs Generate Diverse Molecules? Towards Alignment with Structural Diversity"
_ICLR.cc/2025/Conference — Submitted to ICLR 2025_

### Official Review · Reviewer_xfWi · 2024-10-31

**Soundness:** 2
**Presentation:** 3
**Contribution:** 2
**Rating:** 3
**Confidence:** 4

**Summary:**

This paper tackles an essential problem in AI-driven drug discovery: generating structurally diverse molecules using LLMs. Addressing the challenge that traditional LLMs often produce structurally similar molecules from a single prompt, the authors propose a two-stage approach: (1) supervised fine-tuning to enable autoregressive mutiple molecule generation, and (2) RL to enhance structural diversity. Through experiments with several LLMs and comparison against various decoding schemes, the proposed approach, Div-SFT+RL, demonstrates an improved ability to generate diverse molecular structures, which is crucial for discovering viable drug candidates. The results indicate that Div-SFT+RL outperforms existing models on diversity-related metrics, supporting its potential for applications in drug discovery.

**Strengths:**

1. **Focus on an Essential Problem**: The paper addresses a critical requirement in drug discovery—molecular diversity in generated candidates. By providing a more diverse set of molecules, the proposed method aligns with real-world drug discovery processes, enhancing the chance of identifying successful compounds.
2. **Clear Presentation**: The paper is well-structured, with a clear exposition of the methodology and experimental procedures.
3. **Detailed Experiments**: The authors perform thorough evaluations across multiple metrics and baselines, including the comparison of Div-SFT+RL with various decoding schemes. The use of IntDiv, and NCircles as diversity-related metrics offers a comprehensive view of the method’s performance.

**Weaknesses:**

1. **Limited Discussion of Related Works on RL for Diversity in Molecular Generation**:
   The paper lacks an in-depth discussion of prior reinforcement learning (RL) approaches that also aim to improve molecular diversity, particularly those using RL for targeted or diverse molecule generation. Works such as Blaschke et al. (2020), Pereira et al. (2021), Hu et al. (2023), and He et al. (2024) provide diverse RL strategies in molecular design, which may not all involve LLMs but offer relevant insights on RL methods and should be discussed to position this work better.

   [1] Blaschke, Thomas, et al. "Memory-assisted reinforcement learning for diverse molecular de novo design." Journal of cheminformatics 12.1 (2020): 68.

   [2] Pereira, Tiago, et al. "Diversity oriented deep reinforcement learning for targeted molecule generation." Journal of cheminformatics 13.1 (2021): 21.

   [3] Hu, Xiuyuan, et al. "De novo drug design using reinforcement learning with multiple gpt agents." Advances in Neural Information Processing Systems 36 (2023).

   [4] He, Jiazhen, et al. "Evaluation of reinforcement learning in transformer-based molecular design." Journal of Cheminformatics 16.1 (2024): 95.

2. **Applicability of the ChEBI-20 Dataset for Diversity Evaluation**:
   The use of the ChEBI-20 dataset raises concerns, as each molecular description in this dataset corresponds to a single molecule. Baseline approaches typically focus on generating one molecule to match the target rather than generating a diverse set for a single description. Furthermore, some descriptions may specify molecular details to the extent that producing diverse structures could be contradictory, making ChEBI-20 potentially unsuitable for assessing molecular diversity.

3. **Reward Function’s Balancing of Terms**:
   The reward function includes two terms: one for structural diversity and another for description matching. It would be beneficial to introduce a coefficient to balance these terms, enabling tuning based on the importance of each aspect (diversity vs. adherence to the description).

4. **NCircles Values in Figure 5**:
   In Figure 5, NCircles values at threshold $h=0.75$ are higher than those at $h=0.65$. This seems inconsistent with the metric’s definition, where values at a higher threshold should logically be lower. The unexpected result suggests potential misinterpretation or misuse of the metric, which requires clarification.

**Questions:**

1. **Cluster Representation in Figure 6**:
   Is it reasonable to represent each cluster as a circle in Figure 6? Given that a 2D projection of a circle in a high-dimensional space may not accurately be circular, I wonder if this visualization accurately reflects clustering in chemical space.
2. **Effect of SMILES Randomness on BLEU Scores**:
   Since a single molecule can be represented by multiple SMILES strings, the randomness in SMILES representations could impact BLEU scores in the experiments. Has this potential source of variability been considered, and what effect does it have on the method’s stability or the evaluation of generated molecules?

In conclusion, this paper offers a valuable approach to improving molecular diversity in AI-driven drug discovery, and I will raise my score if my concerns (in Weaknesses and Questions) are well-addressed.

---

> ### Author Response · Authors · 2024-11-22
>
> Dear reviewer xfWi,
>
> We express our deep appreciation for your time and insightful comments. In our updated manuscript, we highlight the changes in $\color{blue}{\text{blue}}$.
>
> In what follows, we address your comments one by one.
>
> ---
>
> **W1. The paper lacks discussion about relevant RL-based methods for diversity in molecular generation.**
>
> Thank you for the valuable suggestion! In **Appendix A.1** of our updated manuscript (also in **third footnote** in **Page 3**), we incorporate discussions on prior RL methods for generating diverse molecules and describe how our work differs from them: they learn a model focused on a fixed target molecular property, whereas our work fine-tunes LLMs to generate diverse molecules given a prompt flexibly describing various target properties.
>
>
> ---
>
> **W2. Applying and evaluating diverse molecular generation on ChEBI-20 datasets may not suitable as a molecular description can be corresponded to a single molecule.**
>
>
> To address this comment, we would like to re-emphasize that the diverse molecular generation aims to increase the probability of the target molecule being included in the generated set of molecules by exploring a wide chemical space. Even for solving problems with one ground truth answer, generating diverse candidates increase probability of including the answer and being informative.
>
>
>
>
> ---
>
> **W3. It would be beneficial to introduce coefficient terms for description-matching and diversity rewards.**
>
> Thanks you for your suggestion! In our updated manuscript, We incorporate the coefficient terms $\lambda_{\text{match}}$ and $\lambda_{\text{div}}$ for description-matching and diversity rewards, respectively.
>
> ---
>
> **W4. In Figure 5, it seems that NCircles with $h=0.65$ and $h=0.75$ metrics do not follow the original metric definition where values at a higher $h$ should logically be lower.**
>
>
> First, we would like to clarify that our NCircles metrics follow the original definition of NCircles metric and are logically correct, but we found that the description of NCircles in our paper is insufficient. We have modified the description of NCircles in our updated manuscript with the following contents.
>
>
> Given a set of molecules $\mathcal{M}$, the NCircles metric computes the size of the largest subset of molecules in which no two molecules are similar to each other. Specifically, it is defined with a Tanimoto similarity $T(\cdot,\cdot)$ and similarity threshold $h$ as follows:
> \begin{equation}
> \text{NCircles}_{h}=\max _{\mathcal{C}\subseteq\mathcal{M}} |\mathcal{C}| \quad \text{s.t. }T(x,y)<h, \forall x \neq y \in \mathcal{C},
> \end{equation}
> where $\mathcal{C}$ is a subset of molecules. Every pair of molecules in $\mathcal{C}$ should have a similarity lower than $h$. Consequently, the value of NCircles metric is increased with respect to increasing $h$ as the condition becomes loose, e.g., $\mathcal{C}=\mathcal{M}$ when $h>1$.
>
> Additionaly, we would like to clarify that this is identical to the definition of NCircles metric in the original paper [1], which is defined with a Tanimoto distance $d(x,y)=1-T(x,y)$:
> \begin{equation}
> \text{NCircles} ^{\text{dist}} _{t} = \max _{\mathcal{C}\subseteq\mathcal{M}} |\mathcal{C}| \quad \text{s.t. } d(x,y)>t, \forall x \neq y \in \mathcal{C},
> \end{equation}
> where $t$ is a distance threshold and $\text{NCircles} ^{\text{dist}} _{t}=\text{NCircles} _{1-h}$.
>
> ---
>
> **Q1. Is it reasonable to represent each cluster as a circle in Figure 6? I wonder if this visualization accurately reflects clustering in chemical space.**
>
> We would like to clarify that the circle in **Figure 6** is just a visualization purpose for explaining the NCircles in 2-D space. Therefore, this visualization does not accurately reflect clustering in chemical space. Specifically, we draw **Figure 6** with the Fruchterman-Reingold force-directed algorithm used in the spring_layout of the NetworkX package [2], where the edge weight is defined with the Tanimoto similarity $T(x,y)$.

---

> ### Author Response · Authors · 2024-11-22
>
> **Q2. Have the authors considered the randomness of SMILES strings in measuring metrics? What effect does the randomness of SMILES have on the stability of method or evaluation?**
>
>
> To remove the randomness stemming from the non-uniqueness of SMILES, i.e., two distinct SMILES strings represent an identical molecule, we applied canonicalization [3]: converts SMILES strings representing an identical molecule into a unique SMILES string. We clarify this in **Line 315** of our updated manuscript.
>
>
>
> ---
>
> [1] Xie et al., How Much Space Has Been Explored? Measuring the Chemical Space Covered by Databases and Machine-Generated Molecules, ICLR 2023
>
> [2] Hagberg et al., Exploring network structure, dynamics, and function using NetworkX, 2008
>
> [3] Weininger et al., SMILES. 2. Algorithm for generation of unique SMILES notation, Journal of chemical information and computer sciences 1989

---

> > ### Comment · Reviewer_xfWi · 2024-11-24
> >
> > Thank you for your detailed response and efforts in addressing my concerns. I appreciate the updates and clarifications provided in the revised manuscript. However, I continue to view **Weakness 2**—the suitability of applying and evaluating diverse molecular generation on the ChEBI-20 dataset—as a significant issue.
> >
> > While I understand the argument that diversity could increase the likelihood of covering the target molecule, I believe that **diversity is not always an appropriate objective for molecule design**. For instance, in tasks like molecule optimization, the goal is to generate new compounds similar to a reference molecule. In such cases, similarity, rather than diversity, is the desired outcome, which is directly contrary to the focus of this paper.
> >
> > Moreover, the key baselines in your paper, **MolT5 and BioT5+**, are designed to generate a single desirable molecule that matches the given description, rather than a diverse set of molecules. This highlights a fundamental mismatch between the baselines’ objectives and your proposed evaluation, which prioritizes diversity. This disconnect raises concerns about the **experimental validity of comparing your method with these baselines**.
> >
> > Therefore, I find the experimental setup of the paper to be unreasonable and not aligned with the goals of diverse molecular generation in the context of the dataset and baselines used. As a result, I do not change my rating.

---

> ### Author Response · Authors · 2024-11-25
>
> Thank you for the response! We would like to clarify the following points to address your comments.
>
> ---
>
> **W2-1. Diversity is not always an appropriate objective as similarity to a reference molecule can be significant rather than diversity, e.g., a molecular optimization task given a reference molecule.**
>
>
> First, we would like to clarify that we have not only considered the diversity but also considered the quality with $r_ {\text{match}}$ which can be defined with the similarity to the reference molecule in the molecular optimization tasks you mentioned. In these tasks, we would like to clarify that both similarity to a reference molecule and diversity are still important, as focusing only on similarity may generate multiple molecules similar to the reference but identical to other generated molecules, i.e., the set of generated molecules may not be informative.
>
> Additionally, in the experiments on the ChEBI-20 dataset, we have already considered a setting similar to the settings mentioned above, where $r_ {\text{match}}$ is computed with the similarity to the target molecule satisfying the given description (as described in **Equation 4** of **Appendix B.2**) to generate diverse molecules similar to the target molecule. We also have considered this in the evaluation (see response of W2-2).
>
> Besides, there exist plenty of significant molecular optimization tasks where diversity is useful, e.g., drug diversification [1,2]. We firmly believe our algorithm will be useful in these scenarios.
>
>
>
> ---
>
>
>
> **W2-2. As MolT5 and BioT5$^{+}$ are designed to generate a single desirable, there are concerns about (1) mismatch between the baselines objectives and your proposed evaluation which prioritizes diversity and (2) experimental validity in comparing your method with MolT5 and BioT5$^{+}$ do not consider the diversity.**
>
> For (1), we would like to clarify that we have also considered the baseline objectives: evaluating how the generated molecules are similar to the target molecules, in defining the set of accepted molecules (as described in **Line 348**) which is also defined as $\mathcal{M}$ to measure $\text{NCircles}$ (as described in **Line 350**), and in defining $\text{Top }10$ scores (as described in **Line 359**), i.e., our comparisons do not only focus on diversity.
>
>
> For (2), we would like to clarify that we compare our method with decoding schemes for diversified generation, e.g., beam search, rather than MolT5 and BioT5$^{+}$ which serve backbones. Our experiments are valid as they aim to show that applying existing decoding schemes is insufficient to obtain diverse and high-quality molecules similar to the target, while our method addresses these limitations.
>
>
> ---
>
> [1] Krantz et al., Diversification of the drug discovery process, Nature Biotechnology 1998
>
> [2] Nippa et al., Enabling late-stage drug diversification by high-throughput experimentation with geometric deep learning, Nature Chemistry 2024

---

### Official Review · Reviewer_oKU9 · 2024-11-01

**Soundness:** 4
**Presentation:** 3
**Contribution:** 3
**Rating:** 8
**Confidence:** 4

**Summary:**

The paper proposes a fine-tuning method to generate diverse molecules using LLMs. The method involves supervised fine-tuning (SFT) and reinforcement learning (RL).

For SFT, the LLM is prompted to generate a molecule with a given property. The prompt is sampled many times, the generated molecules are filtered and then concatenated to create the SFT training set. The SFT step uses a prompt describing the property and requesting a set of molecules. The set of molecules generated previously is appended to this prompt.

For RL, the SFT prompt is used to generate a sequence of molecules. Every time a molecule is generated, the LLM policy is updated using Proximal Policy Optimization (PPO). The reward is based on how well the molecule matches a property and how different it is from the previous molecules.

**Strengths:**

**Originality**
To my knowledge, the method is novel and the problem is under-researched.

**Quality**
The SFT and RL methods are clear and intuitive. The algorithms are well described in the text and code is provided in Supplementary. Figure 4 helps the understanding of the method. The gap in existing LLMs at generating structurally diverse molecules is well explained in the text and in figure 2. The baselines and metrics are mostly well chosen. The results are convincing, with the authors' method producing significant increase in the chosen metrics.

**Clarity** The methods are explained well. The figures are easy to read. Including different texture in Figure 5 helps the readability. The Appendix with all the experimental details and the code in Supplementary is appreciated and helps reproducibility. I also appreciate the title and claims being straightforward and to the point.

**Significance** Generating diverse molecules is important in discovery workflows, thus the problem here is significant. The proposed method is easy to understand and implement. The paper also inspires a few interesting follow-ups and thus I think is an important addition to the community.

**Weaknesses:**

1) In the experiments with BioT5+ and MolT5, the authors used the BLEU score on prompts and molecules from ChEBI-20. I think BLEU is a misleading score in the text to molecule task. For example, in BLEU, the order of words in a sentence matters, while there are many ways of writing a molecules as SMILES strings. Also, small changes in a molecule SMILES in the right place can have high changes in a property, such as in the case of hydrogen bond donors and acceptors. I recommend the authors discuss this in their manuscript and compare it with scores generated by RDKit (which they used in the other experiments).

2) There are a few minor things that could improve the paper:
    - Figure 7 could have different symbols. It was hard to read in a black-and-white version of the paper.
    - Worth considering giving the method a name, this will help wider adoption.

**Questions:**

1) Can you improve the explanation of NCircles? I think I get it, but given it's a relatively new method I yet don't have an intuition for it like I do for things like T-SNE or UMAP. What kinds of hyperparameters does it have? What are the conditions for two nodes to be close to each other? Do the overlap of the circles mean anything or are they a consequence of the force-directed algorithm?

2) In Table 1, you show the outputs of your method for the description "The molecule is a primary aliphatic ammonium ion which is obtained from streptothricin F by protonation of the guanidino and amino groups. It has a role as an antimicrobial agent. It is a guanidinium ion and a primary aliphatic ammonium ion. It is a conjugate acid of a streptothricin F". Are all the generated molecules a conjugate acid of  streptothricin F? Do they have the correct scaffold? This would be important to understand and is related to limitations of the BLEU score.

---

> ### Author Response · Authors · 2024-11-22
>
> Dear reviewer oKU9,
>
> We express our deep appreciation for your time and insightful comments. In our updated manuscript, we highlight the changes in $\color{blue}{\text{blue}}$.
>
> In what follows, we address your comments one by one.
>
> ---
>
> **W1. The authors should (1) discuss the limitation of using BLEU scores in comparing generated molecules and the target molecule satisfying a given description and (2) consider other RDKit-based scores.**
>
>
> We considered BLEU scores following prior studies [1] on text-to-molecule generation. However, we acknowledge their limitations in comparing the generated molecules with the target molecule: they simply measure textual similarity of SMILES strings that may fail to capture the structural or functional information.
>
> To address your concerns, in **Appendix D.2** of our updated manuscript (also in **sixth footnote** in **Page 7**), we (1) discuss the limitations of BLEU scores and (2) conduct additional experiments by replacing BLEU scores with Tanimoto and Dice scores [2] that are more concrete to capture the structural and functional information. One can observe that our fine-tuned model, even though it is trained with the BLEU score, consistently outperforms the baselines in these experiments. We also plan to fine-tune LLMs using Dice scores instead of BLEU scores and include the results in the Appendix.
>
>
>
>
>
> ---
>
> **W2. About minor errors or recommendation.**
>
> Thanks you for valuable suggestions! In our updated manuscript, we modify the symbols in **Figure 7**. We would like to defer naming until the end of the discussion period.
>
>
> ---
>
> **Q1. Can authors improve the explanation of NCircles metric?**
>
>
> To address your comment, we have modified explanation of NCircles metric in **Section 4.1** of our updated manuscript and provided more detailed explanation in **Appendix A.3** of our updated manuscript.
>
>
> To measure the diversity of a given set of molecules $\mathcal{M}$, the NCircles metric computes the size of the largest subset of molecules in which no two molecules are similar to each other. Specifically, this metric is defined with a Tanimoto similarity $T(\cdot,\cdot)$ and a similarity threshold $h$ as follows [3]:
> \begin{equation}
> {{\text{NCircles}}_{h}} = \max _{\mathcal{C}\subseteq\mathcal{M}}  |\mathcal{C}|\quad \text{s.t. }T(x,y)<h, \forall x \neq y \in \mathcal{C},
> \end{equation}
> where $\mathcal{C}$ is a subset of molecules and every pair of molecules in $\mathcal{C}$ should have a similarity lower than $h$. The high NCircles metric implies that the given set of molecules covers a larger volume of molecular space, indicating diversity [3]. We measured this by defining $\mathcal{M}$ as the set of accepted molecules to capture quality and diversity.
>
>
>
> > What kinds of hyperparameters does it have? What are the conditions for two nodes to be close to each other? Do the overlap of the circles mean anything or are they a consequence of the force-directed algorithm?
>
> Therefore, the NCircles metric does not have hyperparameters. Additionally, we guess that the latter two questions stem from **Figure 6** which is just illustrated to explain NCircles in a 2-D space. Here, two nodes are close to each other when they have high similarity, the circle implies the boundary of threshold $h$, and the overlap of circles is just the consequence of the force-directed algorithm.
>
>
>
> ---
>
> **Q2. In Table 1, do the generated molecules correctly follow the given description?**
>
> Unfortunately, it is hard to directly evaluate whether the generated molecules follow the given description due to the complexity of qualitative description. However, it is worth noting that the results in **Figure 7** are measured by evaluating whether the generated molecules exactly follow the molecular quantitative description, e.g., QED value or the number of hydrogen bond donors. This shows how generated molecules from our method correctly follow the given descriptions.
>
>
>
>
> ---
>
> [1] Pei et al., BioT5+: Towards Generalized Biological Understanding with IUPAC Integration and Multi-task Tuning, ACL 2024
>
> [2] Bajusz et al, Why is Tanimoto index an appropriate choice for fingerprint-based similarity calculations?, Journal of Cheminformatics 2015
>
> [3]  Xie et al., How Much Space Has Been Explored? Measuring the Chemical Space Covered by Databases and Machine-Generated Molecules, ICLR 2023

---

> > ### Comment · Reviewer_oKU9 · 2024-11-27
> >
> > Many thanks for addressing my comments. I will keep my score.
> >
> > ---
> >
> > **WP1.**
> > Thank you for adding the additional table. Using Tanimoto and Dice scores is more convincing.
> >
> > **Q1.**
> > Describing NCircles by computing "the size of the largest subset in which no two molecules are similar to each other (Tanimoto similarity below a threshold h)" is clear.
> >
> > **Q2.**
> > Agreed, Figure 7 helps understand the match to description.
> >
> > ---

---

### Official Review · Reviewer_FZz8 · 2024-11-02

**Soundness:** 2
**Presentation:** 2
**Contribution:** 2
**Rating:** 3
**Confidence:** 3

**Summary:**

This paper investigates the limitations of current large language models (LLMs) in generating structurally diverse molecules and proposes a novel two-stage fine-tuning approach to address this challenge. Specifically, the authors apply supervised fine-tuning to enable LLMs to generate a sequence of molecules and subsequently leverage reinforcement learning to enhance structural diversity. Experimental results demonstrate that the proposed method improves molecular diversity compared to existing decoding strategies.

**Strengths:**

This paper identifies a limitation in the molecular generation capabilities of LLMs and proposes a diversity enhancement strategy based on autoregressive generation and reinforcement learning.

**Weaknesses:**

1. The method directly adopts LLMs with reinforcement learning. Thus, the technical novelty of the method is limited.
2. To help readers understand the impact of the approach, the authors could provide additional explanations for the diversity metrics or introduce more similarity evaluation methods.

**Questions:**

1. In Table 3, it is unclear why the comparison is made when the number of generated molecules is different. Additionally, all relevant metrics should be included in the comparison to provide a comprehensive assessment.

2. In Table 4, the authors should specify which large language model (LLM) was used as the basis for fine-tuning and reinforcement learning in the compared methods. Furthermore, when comparing Tables 2 and 4, the performance improvement of the Div-SFT approach appears to be minimal.

---

> ### Author Response · Authors · 2024-11-22
>
> Dear reviewer FZz8,
>
> We express our deep appreciation for your time and insightful comments. In our updated manuscript, we highlight the changes in $\color{blue}{\text{blue}}$.
>
> In what follows, we address your comments one by one.
>
> ---
>
> **W1. The method directly adopts LLMs with reinforcement learning. The technical novelty of the method is limited.**
>
> We would like to clarify that the novelty of our work lies in introducing a new concept: leveraging LLM fine-tuning to generate diverse molecules, rather than simply proposing to adopt LLMs with reinforcement learning. We also note that our work is the first to explore the use of reinforcement learning for generating diverse output with LLMs, which has not been previously addressed in the LLM literature to the best of our knowledge.
>
> ---
>
> **W2. To help readers understand the impact of the approach, can authors provide additional explanations for the diversity metrics or introduce more similarity evaluation methods?**
>
> Thanks you for valuable suggestions! To address your comments, we provide additional explanations for the diversity metrics in **Appendix A.3** of our updated manuscript. We also provide additional explanations of various similarity evaluation methods, e.g., Tanimoto, Dice, and cosine similarities, in **Appendix A.2** of our updated manuscript. Additionally, we update **first footnote** in **Page 1**, **Line 200** in **Section 3**, **fourth footnote** in **Page 4**, and **Line 343** in **Section 4.1** to provide high-level explanations of molecular diversity and similarity.
>
>
>
> ---
>
> **Q1. In Table 3, why is the comparison made when the number of generated molecules differs? Why did authors not include all relevant metrics?**
>
>
>
> The comparison with different numbers of generated molecules aims to show that our method consistently discovers more diverse molecules even when the baseline generates a larger number of molecules.
>
> To address your comment, in our updated manuscript, we include all relevant metrics in **Table 10** of **Appendix D.1**.
>
> ---
>
> **Q2-1. What LLM is used as the basis in Table 4?**
>
> We considered BioT5$^{+}$ in **Table 4**. In our updated manuscript, we clarify this in the captions of **Table 4**.
>
> ---
>
> **Q2-2. The performance improvement of Div-SFT seems to be minor.**
>
> The performance Div-SFT is not appropriate for evaluating the benefits of our algorithm, since it is a subroutine of our algorithm. Instead, one should focus on the result of reinforcement learning (Div-RL), which is our final model. The main purpose of Div-SFT is not to improve the performance, but to repurpose the base LLMs for generating multiple molecules before Div-RL.

---

### Official Review · Reviewer_APHU · 2024-11-04

**Soundness:** 2
**Presentation:** 3
**Contribution:** 2
**Rating:** 3
**Confidence:** 3

**Summary:**

This paper proposes a two-stage fine-tuning approach to address the challenge of generating diverse molecules: (1) supervised fine-tuning to enable LLMs to autoregressively generate molecules in a sequence, and (2) reinforcement learning to enhance structural diversity among the generated molecules.
The contributions of this work include introducing an innovative method that combines supervised learning and reinforcement learning to generate structurally diverse molecules. Additionally, the paper demonstrates that this approach surpasses existing LLMs and traditional decoding methods in producing high-quality, diverse molecular structures.

**Strengths:**

1.This paper addresses an important need in molecular generation by developing a method that enhances structural diversity, a key factor in drug discovery.
2. The two-stage fine-tuning approach, which combines supervised learning with reinforcement learning (RL) for diversity maximization well-designed solution.
3. Through empirical testing, the paper shows that its approach outperforms both current LLM-based molecular generators and advanced decoding schemes, highlighting the practical advantages of its method for achieving high-quality, diverse outputs.
4. The proposed method is adaptable and could be extended to other domains requiring diversity in generated outputs, such as protein design or materials discovery. This versatility increases the impact of the work beyond just molecular generation.

**Weaknesses:**

1. The two-stage fine-tuning approach which is highlighted in this paper is not that novel.
2. Although the model is tested on standard datasets and metrics for structural diversity, it lacks experimental or real-world validation to demonstrate the generated molecules’ practical utility in drug discovery.
3. The proposed method’s multi-stage fine-tuning approach may require extensive hyperparameter tuning and setup, which can hinder reproducibility and adoption among researchers.
4. The paper claim "the first to explore the use of LLMs for generating diverse molecule" and "the first propose a fine-tuning approach for LLMs to generate diverse solution", which is overstate.

**Questions:**

See weakness

---

> ### Author Response · Authors · 2024-11-22
>
> Dear reviewer APHU,
>
> We express our deep appreciation for your time and insightful comments. In our updated manuscript, we highlight the changes in $\color{blue}{\text{blue}}$.
>
> In what follows, we address your comments one by one.
>
> ---
> **W1. The two-stage fine-tuning approach itself is not that novel.**
>
> This is not a weakness of our work, since using the two-stage fine-tuning approach is not our main idea. Our idea to leverage LLMs for diverse molecule generation is novel. We design new reward functions for maximizing the structural diversity of generated outputs. We newly construct experiments and metrics for comparing the diversity of our algorithm and existing LLM decoding algorithms.
>
> ---
>
> **W2. Although tested on standard datasets and metrics, this works lacks experimental or real-world validation of the practical utility of the generated molecules.**
>
> Evaluating on the standard datasets and metrics is sufficient, since the datasets and metrics were designed to be representative of the practical utility of the generated molecules [1,2,3]. Our belief aligns with the vast literature of computational methods for designing new molecules [4,5,6,7].
>
> ---
>
> **W3. The proposed approach may require extensive setup or hyperparameter tuning, which can hinder future adoption and reproducibility.**
>
> We clarify that our approach requires minimal setup and hyperparameter tuning due to following the widely used two-stage fine-tuning approach. Our work requires a similar number of hyperparameters compared to many of the two-stage fine-tuning approaches. We ensure reproducibility by providing the code in the **Supplementary Materials**, specifying hyperparameter values in **Appendix B**. We also plan to release the full parameters of the fine-tuned models once our work is accepted.
>
> ---
>
> **W4. The authors overstate that (1) "the first to explore the use of LLMs for generating diverse molecule" and (2) "the first propose a fine-tuning approach for LLMs to generate diverse solution".**
>
> We do not think this is an overstate, since we were unable to find any works that claim (1) and (2). We will be happy to tone down our claims if you could provide any reference that already claims (1) and (2).
>
> ---
>
> [1] Edwards et al., Text2Mol: Cross-Modal Molecule Retrieval with Natural Language Queries, EMNLP 2021
>
> [2] Polykovskiy et al., Molecular Sets (MOSES): A Benchmarking Platform for Molecular Generation Models, Frontiers in Pharmacology 2020
>
> [3] Xie et al., How Much Space Has Been Explored? Measuring the Chemical Space Covered by Databases and Machine-Generated Molecules, ICLR 2023
>
> [4] Hu et al. De novo drug design using reinforcement learning with multiple gpt agents, NeurIPS 2023
>
> [5] Pei et al., BioT5+: Towards Generalized Biological Understanding with IUPAC Integration and Multi-task Tuning, ACL 2024
>
> [6] Yu et al., LlaSMol: Advancing Large Language Models for Chemistry with a Large-Scale, Comprehensive, High-Quality Instruction Tuning Dataset, COLM 2024
>
> [7] He et al. Evaluation of reinforcement learning in transformer-based molecular design, Journal of Cheminformatics 2024

---

> ### Comment · Reviewer_APHU · 2024-11-26
>
> Thank you for your detailed response! While it addressed some of my concerns, I still have the following questions:
>
> 1. "the first to explore the use of LLMs for generating diverse molecule"
> Your work focuses on using reinforcement learning (RL) to redesign the reward function and improve diversity, rather than leveraging Large Language Models (LLMs) themselves to generate diverse molecules. The pretraining stage didn't enhance diversity and the pertaining approach is not novel either. RL fine-tuning play the role of improve diversity.
>
> 2. "the first propose a fine-tuning approach for LLMs to generate diverse solution"
> Your proposal to fine-tune LLMs is not that novel, as similar approaches have been explored before (e.g., Reinvent [1]),(they didn't use your redesigned reward function). Your method uses the PPO algorithm, and the multiple stages can be regarded as  multiple epochs in traditional RL. Therefore, novelty leads to diversity still relies on the redesigned reward function, but it's quite limited.
>
> Therefore, I will keep my score.
>
> [1]Reinvent 4: Modern AI–driven generative molecule design

---

> > ### Author Response · Authors · 2024-11-27
> >
> > Thank you for your response. It seems that you are still concerned that we are over-claiming our contributions. We further provide our response.
> >
> > ---
> >
> > **W4-1. "the first to explore the use of LLMs for generating diverse molecule" Your work focuses on using RL to improve diversity, rather than leveraging LLMs to generate diverse molecules.**
> >
> > Our work explicitly focuses on the idea of leveraging the remarkable expressive power of LLMs to (1) understand the textual conditions and (2) capture the long context for sequentially generating molecules. We use the off-the-shelf RL algorithms since this is not our main focus.
> >
> > ---
> >
> > **W4-2. "the first propose a fine-tuning approach for LLMs to generate diverse solution" Your proposal to fine-tune LLMs is not that novel, as similar approaches have been explored before (e.g., Reinvent [1]).**
> >
> > We would like to clarify that Reinvent [1] did not use LLMs and just chose transformer-based model architectures. We also note that we propose to autoregressively generate the molecules in a concatenated sequence, exploiting the power of LLMs, which was not considered in previous works (including [1]).
> >
> >
> > ---
> >
> > [1] Loeffler et al. Reinvent 4: Modern AI–driven generative molecule design, Journal of Cheminformatics 2024.

---

### Meta-Review · Area_Chair_KboK · 2024-12-22

**Metareview:**

In this work, authors propose a two-stage fine-tuning approach for large language models (LLMs) to generate diverse molecular structures for drug discovery applications. The method combines supervised fine-tuning to enable autoregressive molecule generation followed by reinforcement learning optimization to maximize structural diversity while maintaining adherence to target properties. The authors evaluate their approach against existing LLMs and decoding schemes using various diversity metrics.

There are several strengths for the work as follows. The paper addresses an important need in computational drug discovery by focusing on molecular diversity generation. The methodology is clearly presented with detailed experimental protocols and comprehensive evaluations across multiple metrics (Reviewer oKU9). The authors provide thorough empirical validation showing improvements over baseline approaches in generating structurally diverse molecules that match target properties. The supplementary materials include code and detailed experimental settings that support reproducibility.

However, there are several limitations for the work as evident from the discussion during the review period. The two-stage fine-tuning approach largely adopts existing methods without substantial innovation (Reviewer APHU, FZz8). While the authors argue their contribution lies in applying these techniques to molecular diversity generation, the core technical components are standard. Reviewer xfWi raises significant issues about the appropriateness of using ChEBI-20 for diversity evaluation, since it maps descriptions to single target molecules. While the authors argue diversity remains valuable even with single targets, this creates a fundamental mismatch between the baseline models' objectives and the proposed evaluation metrics. Multiple reviewers (APHU, FZz8) note that the paper's claims of being ``first'' to explore LLMs for diverse molecule generation are overstated, given existing work in this space. Further, the reliance on BLEU scores for molecular similarity is problematic given the multiple valid SMILES representations for identical molecules (Reviewer oKU9). While the authors added Tanimoto and Dice scores in revision, this highlights initial gaps in the evaluation approach.

Based on these comments, the manuscript cannot be recommended for acceptance at this point. To strengthen future submissions, the authors should: (1) develop a more appropriate evaluation framework that fairly compares diversity-oriented approaches, (2) more precisely position their technical contributions relative to existing work, and (3) strengthen the novelty of their methodological approach beyond standard fine-tuning techniques. More comments below.

**Additional Comments On Reviewer Discussion:**

During the discussion, the following important points were noted.

1. On technical novelty, reviewer APHU initially questioned the innovation of the two-stage fine-tuning approach and stated that the paper over-claimed its contributions. The authors responded by clarifying that their novelty lies not in the two-stage approach itself, but in leveraging LLMs specifically for diverse molecule generation. However, APHU remained unconvinced, noting that the diversity improvements stem primarily from the reward function redesign rather than novel use of LLMs.

2. Reviewer FZz8 raised concerns about the limited performance improvement of the Div-SFT approach and requested clarification about which LLM was used as the baseline. The authors addressed this by explaining that Div-SFT is merely a subroutine and not the final model - the focus should be on Div-RL results. They also clarified that BioT5+ was used as the baseline.

3. Reviewer oKU9 highlighted issues with using BLEU scores for molecular similarity evaluation and requested better explanation of the NCircles metric. The authors acknowledged these limitations and added Tanimoto and Dice scores in their revision. They also provided a clearer mathematical formulation of NCircles. oKU9 was satisfied with these additions and maintained their positive assessment.

4. The most critical discussion emerged from reviewer xfWi's concerns about the fundamental mismatch between evaluation objectives and baseline models when using the ChEBI-20 dataset. While the authors argued that diversity remains valuable even for single-target problems, xfWi maintained that this creates an unfair comparison since baselines like MolT5 and BioT5+ were designed for single-molecule generation. The authors' final response attempted to justify their evaluation framework by emphasizing that they compared against diverse decoding schemes rather than the base models, but this did not fully address the core experimental design concern.

In weighing these discussions, I found that while the authors made good efforts to address many technical clarifications and evaluation metrics, the fundamental concerns about experimental validity raised by xfWi and the limited technical novelty highlighted by APHU remained inadequately resolved. These unresolved issues, combined with the over-claimed contributions noted in multiple reviews, ultimately support the rejection decision despite the authors' detailed responses during the rebuttal period.

The discussion reveals that while the authors were responsive and made several improvements to their manuscript, the core issues affecting the paper's suitability for ICLR publication - experimental design validity and technical innovation - were not sufficiently addressed through the rebuttal process.

---

### Decision · Program_Chairs · 2025-01-22

Reject